# TI-MAE: SELF-SUPERVISED MASKED TIME SERIES AUTOENCODERS

## ABSTRACT

Multivariate Time Series forecasting has been an increasingly popular topic in various applications and scenarios. Recently, contrastive learning and Transformer-based models have achieved good performance in many long-term series forecasting tasks. However, there are still several issues in existing methods. First, the training paradigm of contrastive learning and downstream prediction tasks are inconsistent, leading to inaccurate prediction results. Second, existing Transformer-based models which resort to similar patterns in historical time series data for predicting future values generally induce severe distribution shift problems, and do not fully leverage the sequence information compared to self-supervised methods. To address these issues, we propose a novel framework named Ti-MAE, in which the input time series are assumed to follow an integrate distribution. In detail, Ti-MAE randomly masks out embedded time series data and learns an autoencoder to reconstruct them at the point-level. Ti-MAE adopts mask modeling (rather than contrastive learning) as the auxiliary task and bridges the connection between existing representation learning and generative Transformer-based methods, reducing the difference between upstream and downstream forecasting tasks while maintaining the utilization of original time series data. Experiments on several public real-world datasets demonstrate that our framework of masked autoencoding could learn strong representations directly from the raw data, yielding better performance in time series forecasting and classification tasks. The code will be made public after this paper is accepted.

## 1 INTRODUCTION

Time series modeling has an urgent need in many fields, such as time series classification (Dau et al., 2019), demand forecasting (Carbonneau et al., 2008), and anomaly detection (Laptev et al., 2017). Recently, long sequence time series forecasting (LSTF), which aims to predict the change of values in a long future period, has aroused significant interests of researchers. In the previous work, most of the self-supervised representation learning methods on time series aim to learn transformation-invariant features via contrastive learning to be applied on downstream tasks. Although these methods perform well on classification tasks, there is still a gap between their performance and other supervised models on forecasting tasks. Apart from the inevitable distortion to time series caused by augmentation strategies they have borrowed from vision or language, the inconsistency between upstream contrastive learning approaches and downstream forecasting tasks should be also a major cause of this problem. Besides, as the latest contrastive learning frameworks (Yue et al., 2022; Woo et al., 2022a) reported, Transformer (Vaswani et al., 2017) performs worse than CNN-based backbones, which is also not consistent with our experience. We have to reveal the differences and relationships between existing contrastive learning and supervised methods on time series.

As an alternative of contrastive learning, denoising autoencoders (Vincent et al., 2008) are also used to be an auxiliary task to learn intermediate representation from the data. Due to the ability of Transformer to capture long-range dependencies, many of existing methods (Zhou et al., 2021; Wu et al., 2021; Woo et al., 2022b) focused on reducing the time complexity and memory usage caused by vanilla attention mechanism such as sparse attention or correlation to process longer time series. These transformer-based models all follow the same training paradigm as Figure 1a shows, which learns similar patterns from input historical time series segments and predict future time series values

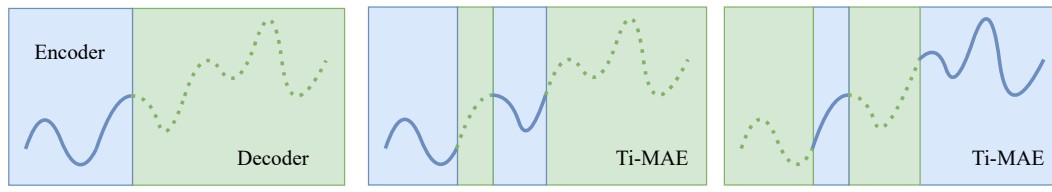

(a) End-to-end forecasting.      (b) Random masking applied in Ti-MAE.

Figure 1: Different masking strategies in generative Transformer-based models on time series, where blue areas signify the sequence fed into the encoder and green areas means the sequence to be generated. **Left**: The training paradigm of existing Transformer-based forecasting models, which can be seen as a special continuous masking strategy (only masks future time series and reconstructs them). **Right**: Random masking strategy applied in Ti-MAE, which can produce different views fed into the encoder in each iteration, fully leveraging the whole input time series.

from captured patterns. These so-called generative Transformer-based models are actually a special kind of denoising autoencoders, where we only mask the future values and reconstruct them.

However, this continuous masking strategy is usually accompanied by two severe problems. For one thing, continuous masking strategy will limit the learning ability of the model, which captures only the information of the visible sequence and some mapping relationship between the historical and the future segments. Similar problems have been reported in vision tasks (Zhang et al., 2017). For another, continuous masking strategy will induce severe distribution shift problems, especially when the prediction horizon is longer than input sequence. In reality, most of the time series data collected from real scenarios are non-stationary, whose mean or variance changes over time. Similar problems were also observed in previous studies (Qiu et al., 2018; Oreshkin et al., 2020; Wu et al., 2021; Woo et al., 2022a). Most of them have tried to disentangle the input time series into a trend part and a seasonality part in order to enhance the capture of periodic features and to make the model robust to outlier noises. Specifically, they utilize moving average implemented by one average pooling layer with a fixed size sliding window to gain trend information of input time series. Then they capture

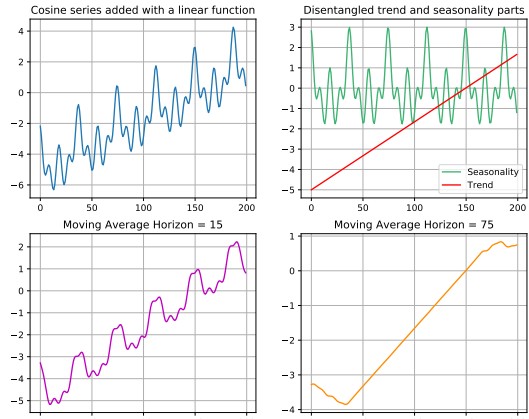

Figure 2: Example of disentanglement. **Top Left**: Simulated input cosine series added with a linear trend. **Top Right**: The true trend and seasonality parts of the input. **Bottom Left**: Disentangled trend part though average pooling with the sliding window size of 15. **Bottom Right**: Disentangled trend part though average pooling with the sliding window size of 75.

seasonality features from periodic sequences, which are obtained by simply subtracting trend items from the original signal. To further clarify the mechanism of this disentanglement, we intuitively propose an easy but comprehensible description of disentangled time series as

$$\boldsymbol{y}(t) = \textit{Trend}(t) + \textit{Seasonality}(t) + \text{Noises}. \quad (1)$$

For better illustration, we simply use polynomial series $\sum_n t^n$ and Fourier cosine series $\sum_n \cos^n t$ to respectively describe trend parts and seasonality parts of the original time series in Eq.(1). Apparently, the seasonality part is stationary when we set a proper observation horizon (not less than the maximum period of seasonality parts), while the moments of the trend part change continuously over time. Figure 2 illustrates that the size of sliding window in average pooling layer plays a vital role in the quality of disentangled trend part. Natural time series data generally have more complex periodic patterns, which means we have to employ longer sliding windows or other hierarchical disposals. In addition, when moving average is used to capture the trend parts, both ends of a sequence need to be

padded for alignment, which causes inevitable data distortion at the head and tail. These observed phenomenons suggest there are still some unresolved issues in the current disentanglement.

To address these issues, this paper proposes a novel Transformer-based framework named Ti-MAE as shown in Figure 3. Ti-MAE randomly masks out parts of embedded time series data and learns an autoencoder to reconstruct them at the point-level in the training stage. Figure 1 shows the difference between random masking and fixed continuous masking in end-to-end models, where we adequately leverage all the input sequence with different combination of visible tokens. Random masking takes the overall distribution of inputs into consideration, which can therefore alleviate the distribution shift problem. Moreover, with the power of pre-training or representation learning embodied in the encoder-decoder structure, Ti-MAE provides a universal scheme for both forecasting and classification. The contributions of our work are summarized as follows:

- We provide a novel perspective to bridge the connection between existing contrastive learning and generative Transformer-based models on time series and point out the inconsistency and deficiencies of them on downstream tasks.

- We propose Ti-MAE, a masked time series autoencoders which can learn strong representations with less inductive bias or hierarchical trick. Masking time series modeling in training stage adequately leverages the input data and successfully alleviates the distribution shift problem. Due to the flexible setting of masking ratio, Ti-MAE can adapt to complex scenarios which require the trained model to make forecasting simultaneously for multiple time windows with various sizes without re-training.

- Ti-MAE has achieved excellent performance for both forecasting and classification tasks on several public real-world time series datasets, demonstrating the power of pre-training or representation learning of Ti-MAE in the time series domain.

## 2    RELATED WORK

### 2.1    TRANSFORMER-BASED TIME SERIES MODEL

Due to the ability of Transformer to capture long-range dependencies, Transformer-based model has been widely used in language and vision tasks. Song et al. (2018); Ma et al. (2019); LI et al. (2019) tried to directly apply vanilla Transformer to time series data but failed in long sequence time series forecasting tasks as self-attention operation scales quadratically with the input sequence length. Child et al. (2019); Zhou et al. (2021); Liu et al. (2022) noticed the long tail distribution in self-attention feature map so that they utilized sparse attention mechanism to reduce time complexity and memory usage of vanilla Transformer for processing longer sequences. Unfortunately, applying too long input sequence in training stage will degrade the forecasting accuracy of the model (Wu et al., 2021), which is in contrast to the ability that Transformer-based model can capture long-range dependencies. Some of the latest works like ETSformer (Woo et al., 2022b) and FEDformer (Zhou et al., 2022) also rely heavily on disentanglement and extra introduced domain knowledge.

### 2.2    TIME SERIES REPRESENTATION LEARNING

Self-supervised representation learning has achieved good performance in time series domain, especially using contrastive learning to learn a good intermediate representation. Lei et al. (2019); Franceschi et al. (2019) used loss function of metric learning to preserve pairwise similarities in the time domain. CPC (van den Oord et al., 2018) first proposed contrastive predictive coding and InfoNCE, which treats the data from the same sequence as positive pairs while the different noise data from the mini-batch as negative pairs. Different data augmentations on time series data were proposed to capture transformation-invariant features at semantic level (Eldele et al., 2021; Yue et al., 2022). CoST (Woo et al., 2022a) introduced extra inductive biases in frequency domain through DFT and separately processed disentangled trend and seasonality parts of the original time series data to encourage discriminative seasonal and trend representations. Almost all of these methods rely on heavily data augmentation or other domain knowledge like hierarchy and disentanglement.

## 2.3 MASKED DATA MODELING

Masked language modeling is a widely adapted method for pre-training in NLP. BERT (Devlin et al., 2019) holds out a portion of the input sequence and predicts the missing content in training stage, which can generate good representations to various downstream tasks. Masked image encoding methods are also used for learning image representations. Pathak et al. (2016) recovered a small portion of missing regions using convolution. Motivated by the huge successes in NLP, recent methods (Bao et al., 2021; Dosovitskiy et al., 2021) are resort to Transformers to predict unknown pixels. MAE (He et al., 2021) proposed to mask a high portion of patches and retain a small set of visible patches received by encoder in pre-training on image data. Zerveas et al. (2021) directly masked a small portion of time series to learn representations. Pang et al. (2022); Tong et al. (2022); Feichten-hofer et al. (2022) have shown that MAE-style methods are effective to learn good representations. Specially designed for enhancing the performance of spatial-temporal graph neural networks, TS-former (Shao et al., 2022) concurrently tried to use MAE to generate intermediate representations for spatial-temporal data. In addition, ExtraMAE (fang Zha, 2022) used RNN as the backbone with the masking scheme for fast time series generation. Different from these methods, Ti-MAE inherits the advantages of Transformer in modeling long time dependencies, and develops a universal time series generation method for both forecasting and classification, with outstanding performance in comparison with state-of-the-art transformer time series models and contrastive learning methods.

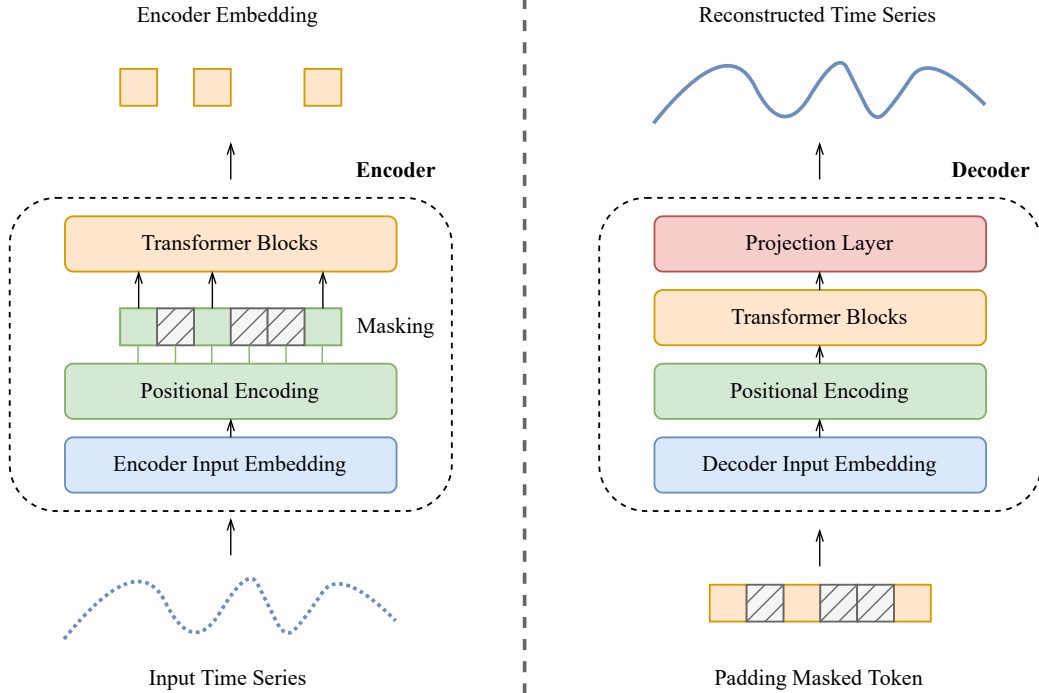

Figure 3: **Ti-MAE structure overview**. **Left**: The encoder receives raw time series inputs. After embedding inputs into tokens on timestamp, we randomly mask a large subset of tokens. Then we feed all the visible tokens into Transformer blocks to capture dependencies. **Right**: The lighter decoder processes encoded tokens padded with masked tokens and reconstructs the original time series at the point-level.

## 3 METHODOLOGY

### 3.1 PROBLEM DEFINITION

Let $\mathcal{X} = (\boldsymbol{x}_1, \boldsymbol{x}_2, \ldots, \boldsymbol{x}_T) \in \mathbb{R}^{T \times m}$ be a multivariate time series instance with length of $T$, where $m$ is the dimension of each signal. Given a historical multivariate time series segment $\mathcal{X}_h \in \mathbb{R}^{h \times m}$ with length of $h$, forecasting tasks aim to predict the next $k$ steps values of $\mathcal{X}_f \in \mathbb{R}^{k \times n}$ where

$n \le m$. For classification tasks, we should match the categorical ground truth from a set of labels $\mathcal{C}$ and each time series instance $\mathcal{X}$.

## 3.2 MODEL ARCHITECTURE

The overall architecture of Ti-MAE is shown in Figure 3. Similar as all autoencoders, our framework has an encoder that maps the observed time series signal $\mathcal{X} \in \mathbb{R}^{T \times m}$ to a latent representation $\mathcal{H} \in \mathbb{R}^{T \times n}$, and a decoder that reconstructs the original sequence from the embedding of the encoder on timestamp. Motivated by the great success of other MAE-style approaches (He et al., 2021; Feichtenhofer et al., 2022; Hou et al., 2022), we also adopt an asymmetric design that the encoder only operates visible tokens after applying masking on input embedding, and a lighter decoder processes encoded tokens padded with masked tokens and reconstructs the original time series at the point-level. More details of each component are introduced as follows.

**Input embedding.** Unlike other time series modeling methods, we have not adopted any multi-scale or complex convolution scheme like dilated convolution. Given a time series segment, we directly use one 1-D convolutional layer to extract local temporal features on timestamp across channels. Fixed sinusoidal positional embeddings are added to maintain the position information. Be different from other temporal data embedding approaches, we do not add any handcrafting task-specific or date-specific embeddings so as to introduce as little inductive bias as possible.

**Masking.** After tokenizing original temporal data into tokens on timestamp, we randomly sample a subset of tokens without replacement which follows the uniform distribution and mask the remaining parts. It is hypothesized and summarized in (He et al., 2021; Feichtenhofer et al., 2022) that the masking ratio is related to the information density and redundancy of the data, which has an immense impact on the performance of the autoencoders. Generally speaking, natural language has higher information density due to its highly discrete word dis-

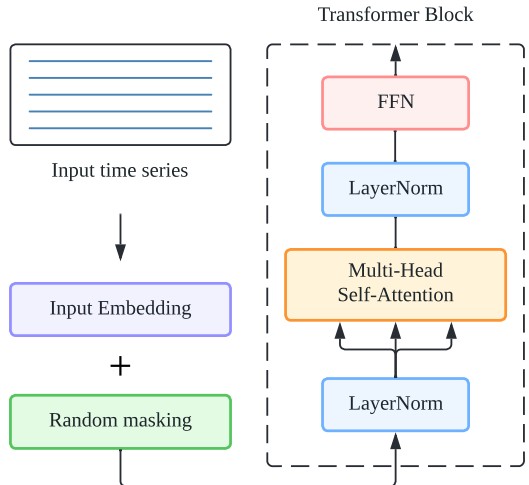

Figure 4: Ti-MAE encoder overview. **Left**: Encoder input embedding. **Right**: Details of one Transformer block used in both Ti-MAE encoder and decoder, where we utilize pre-norm instead of post-norm scheme.

tribution, while images are of heavy spatial redundancy. Specifically, single pixel in one image has lower semantic information so that we can reconstruct a missing region from neighboring pixels by interpolation with little understanding of contents. Thus, data with lower information density should be applied a higher masking ratio to largely eliminate redundancy and prevent the model from focusing only on low-level semantic information. As a benchmark model often used in natural language, BERT (Devlin et al., 2019) uses a masking ratio of 15% while MAE uses a ratio of 75% for images (He et al., 2021) and 90% for videos (Feichtenhofer et al., 2022). Similar as images, time series data also have local continuity so that we should determine a high masking ratio in training stage. The optimal masking ratio of multivariate time series we observe is also around 75%.

**Ti-MAE Encoder.** Our encoder is a set of vanilla Transformer blocks with input embedding but utilizes pre-norm instead of post-norm in each block, which is shown as Figure 4. Like other MAE-style methods, Ti-MAE's encoder is applied only on visible tokens after embedding and random masking. This design significantly reduces time complexity and memory usage compared to full encoding.

**Ti-MAE Decoder.** Our decoder also contains a set of vanilla Transformer blocks applied on the union of the encoded visible tokens and learnable randomly initialized mask tokens. Following (He et al., 2021), the decoder is designed to be smaller than the encoder. Notably, we add positional embeddings to all tokens after padding to supplement the location information of the missing parts.

The last layer of the decoder is a linear projection layer which reconstructs the input by predicting all the values at the point-level. The training loss function is the mean squared error (MSE) between the original time series data and the prediction over masking regions.

The encoder and decoder of Ti-MAE are both agnostic to the sequential data with as less domain knowledge as possible. There is no date-specific embedding, hierarchy or disentanglement in contrast to other architectures (Zhou et al., 2021; Wu et al., 2021; Yue et al., 2022; Woo et al., 2022a). Compared to masked autoencoders used in vision tasks, a lot of parameter settings of Ti-MAE have been adjusted to better fit the time series data. We keep the point-level modeling rather than patch embedding for the consistency between masked modeling and downstream forecasting tasks. Unlike (Shao et al., 2022), we directly generate future values from the decoder as prediction, maintaining the consistency of training and inference stages.

## 4 EXPERIMENTS

### 4.1 EXPERIMENTAL SETUP

**Datasets.** We conduct extensively experiments on several public real-world datasets, covering time series forecasting and classification applications. (1) ETT (Electricity Transformer Temperature) (Zhou et al., 2021) consists of the data collected from electricity transformers, recording six power load features and oil temperature. (2) Weather[1] contains 21 meteorological indicators like humidity, pressure in 2020 year from nearly 1600 locations in the U.S.. (3) Exchange (Lai et al., 2018) is a collection of exchange rates among eight different countries from 1990 to 2016. (4) ILI[2] records the weekly influenza-like illness (ILI) patients data from Centers for Disease Control and Prevention of the United States between 2002 and 2021, describing the ratio of patients observed with ILI and the total number of patients. (5) The UCR archive (Dau et al., 2019) has 128 different datasets covering multiple domains like object outlines, traffic and body posture. We follow the same protocol and split all forecasting datasets into training, validation and test set by the ratio of 6:2:2 for the ETT dataset and 7:1:2 for other datasets. For classification, each dataset of UCR archive has been already divided into training and test set where the size of test set is greatly larger than training set in order to be accord with the actual scenarios.

**Baselines.** We select two types of baselines, Transformer-based end-to-end and representation learning methods which have public official codes. For time series forecasting tasks, we select four latest state-of-the-art representation learning models: CoST (Woo et al., 2022a), TS2Vec (Yue et al., 2022), TNC (Tonekaboni et al., 2021) and MoCo (Chen et al., 2021) applied on time series and four Transformer-based end-to-end models: FEDformer (Zhou et al., 2022), ETSformer (Woo et al., 2022b), Autoformer (Wu et al., 2021) and Informer (Zhou et al., 2021). For time series classification tasks, we include more competitive unsupervised representation learning methods: TS2Vec, T-Loss (Franceschi et al., 2019), TS-TCC (Eldele et al., 2021), TST (Zerveas et al., 2021), TNC (Tonekaboni et al., 2021) and DTW (Chen et al., 2013).

**Implementation Details.** The encoder and decoder of Ti-MAE both use 2 layers of vanilla Transformer blocks with 4 heads self-attention. The number of hidden states dimension is set to 64, which is significantly lower than other existing methods (e.g., 320, 512). Ti-MAE is trained with MSE loss, using the Adam optimizer (Kingma & Ba, 2015) with an initial learning rate of $1e-3$. We apply a batch size of 64 and sampling time of 30 in each iteration. We use mean squared error (MSE) $\frac{1}{n}\sum_{i=1}^{n}(\boldsymbol{y}-\hat{\boldsymbol{y}})^2$ and mean absolute error (MAE) $\frac{1}{n}\sum_{i=1}^{n}|\boldsymbol{y}-\hat{\boldsymbol{y}}|$ as evaluation metrics on forecasting tasks, and average accuracy with critical difference (CD) on classification tasks. All the models are implemented in PyTorch (Paszke et al., 2019) and trained/tested on a single Nvidia V100 32GB GPU.

### 4.2 TIME SERIES FORECASTING

To simulate different forecasting scenarios, we evaluate models under different future horizons, covering short-term and long-term forecasting cases. Tables 1 and 2 summarize the multivariate time series forecasting evaluation results of four datasets.

---

[1]https://www.ncei.noaa.gov/data/local-climatological-data/

[2]https://gis.cdc.gov/grasp/fluview/fluportaldashboard.html

Table 1: Multivariate time series forecasting results compared to representation learning methods.

| Method | | Ti-MAE | | CoST | | TS2Vec | | TNC | | MoCo | |
|---|---|---|---|---|---|---|---|---|---|---|---|
| Metric | | MSE | MAE | MSE | MAE | MSE | MAE | MSE | MAE | MSE | MAE |
| ETTh | 12 | **0.2629** | **0.3462** | 0.3374 | 0.4001 | 0.5817 | 0.5217 | 0.6056 | 0.5389 | 0.6419 | 0.5518 |
| | 24 | **0.3520** | **0.3924** | 0.3832 | 0.4301 | 0.5897 | 0.5312 | 0.6331 | 0.5616 | 0.6491 | 0.5630 |
| | 48 | **0.3977** | **0.4173** | 0.4342 | 0.4665 | 0.6242 | 0.5545 | 0.6934 | 0.6001 | 0.6804 | 0.5867 |
| | 96 | **0.4266** | **0.4301** | 0.5229 | 0.5201 | 0.6812 | 0.5699 | 0.7538 | 0.6391 | 0.7618 | 0.6323 |
| | 128 | **0.4493** | **0.4436** | 0.5742 | 0.5529 | 0.7190 | 0.5919 | 0.7949 | 0.6622 | 0.8062 | 0.6577 |
| | 168 | **0.5091** | **0.4594** | 0.6326 | 0.5838 | 0.7621 | 0.6387 | 0.8360 | 0.6849 | 0.8201 | 0.6742 |
| Weather | 12 | **0.0932** | **0.1460** | 0.1652 | 0.2630 | 0.1481 | 0.2367 | 0.1819 | 0.2508 | 0.1642 | 0.2521 |
| | 24 | **0.1226** | **0.1809** | 0.2719 | 0.3525 | 0.3011 | 0.3551 | 0.3118 | 0.3731 | 0.3112 | 0.3651 |
| | 48 | **0.1633** | **0.2280** | 0.3662 | 0.3672 | 0.3741 | 0.4178 | 0.3803 | 0.4117 | 0.3717 | 0.4163 |
| | 96 | **0.2123** | **0.2735** | 0.4119 | 0.4266 | 0.4289 | 0.4507 | 0.4176 | 0.4174 | 0.4077 | 0.4419 |
| | 128 | **0.2197** | **0.2805** | 0.4302 | 0.4686 | 0.4663 | 0.4839 | 0.4569 | 0.4824 | 0.4582 | 0.4693 |
| | 168 | **0.2460** | **0.3049** | 0.4636 | 0.4914 | 0.4909 | 0.5061 | 0.4789 | 0.4950 | 0.4820 | 0.4992 |
| Exchange | 24 | **0.0697** | **0.1889** | 0.1365 | 0.2721 | 0.0873 | 0.2245 | 0.0834 | 0.2084 | 0.1058 | 0.2553 |
| | 48 | **0.1255** | **0.2448** | 0.2532 | 0.3783 | 0.1666 | 0.3047 | 0.1648 | 0.2928 | 0.2018 | 0.3588 |
| | 96 | **0.1701** | **0.2972** | 0.5408 | 0.5645 | 0.4686 | 0.5098 | 0.3756 | 0.4510 | 0.4162 | 0.5002 |
| | 128 | **0.2208** | **0.3242** | 0.6786 | 0.6334 | 0.6540 | 0.6036 | 0.5483 | 0.5441 | 0.5950 | 0.6050 |
| | 168 | **0.2151** | **0.3316** | 0.8859 | 0.7338 | 0.9683 | 0.7348 | 0.7701 | 0.6470 | 0.8079 | 0.6997 |
| | 196 | **0.2123** | **0.3291** | 0.9720 | 0.7703 | 1.1692 | 0.8084 | 0.9495 | 0.7204 | 0.9534 | 0.7591 |
| ILI | 24 | **2.7474** | 1.0740 | 2.8332 | **1.0656** | 3.5111 | 1.1882 | 3.3729 | 1.2011 | 2.9399 | 1.1014 |
| | 36 | **2.7124** | **1.0348** | 3.1439 | 1.1197 | 3.7813 | 1.2588 | 4.0722 | 1.3292 | 3.4974 | 1.2212 |
| | 48 | **2.6138** | **1.0399** | 3.4153 | 1.1725 | 4.1892 | 1.3319 | 4.1239 | 1.3239 | 3.7872 | 1.2713 |
| | 60 | **2.2889** | **0.8940** | 3.7917 | 1.2553 | 4.2588 | 1.3352 | 3.9937 | 1.3063 | 3.8137 | 1.2695 |
| | 72 | **2.0820** | **0.8372** | 4.0823 | 1.3232 | 4.1868 | 1.3431 | 4.0423 | 1.3294 | 3.8818 | 1.3023 |
| | 96 | **2.4419** | **1.0287** | 4.2442 | 1.3755 | 4.3677 | 1.3756 | 4.2162 | 1.3594 | 4.2148 | 1.3530 |

In Table 1, Ti-MAE consistently improves the performance in across all datasets of different prediction horizons. Specifically, Ti-MAE achieves a MAE decrease of 15.7% in ETT, 42.3% in Weather, 45.5% in Exchange and 19.2% in ILI compared to representation learning frameworks. Notably, our Ti-MAE does not require any extra regressor after pre-trained because its decoder can directly generate future time series to be predicted given the input sequence and masking ratio. In Table 2, Ti-MAE († indicates fine-tuned version) also shows more compatible performance compared to other Transformer-based end-to-end supervised methods. It must be stressed that we have pre-trained **only one** Ti-MAE model while all the end-to-end supervised models should be trained separately for different settings. Then we

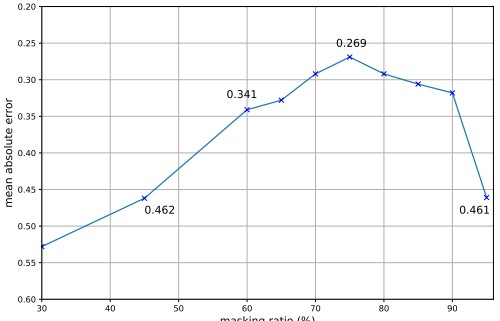

Figure 5: The optimal masking ratio is around 75%. Lower or higher masking ratio will degrade the performance of prediction.

just utilize its encoder (parameters have been frozen) with an additional linear projection layer for fine-tuning at different prediction horizons. Runtime analysis comapred to other Transformer-based models could be seen at appendix. To further explore the impact of main properties of Ti-MAE, we conduct extensive ablation experiments on Weather under input sequence length of 200 and prediction horizon of 100 setting for evaluation. Table 3 demonstrates all the results of ablation study.

**Masking ratio.** Figure 5 and Table 3a show the influence of the masking ratio. The optimal ratios are around 75%, which is in contrast to BERT (Devlin et al., 2019) and video-MAE (Feichtenhofer et al., 2022) but similar to MAE for images (He et al., 2021). The high masking ratio induces the model to process fewer tokens and learn high-level semantic information. We can see that lower masking ratios perform worse even if the encoder could see more tokens because the model trained with lower masking ratio may simply recover the values by interpolation or extrapolation, focusing on low level semantic features locally.

Table 2: Multivariate time series forecasting results compared to end-to-end methods.

| Method | | Ti-MAE† | | ETSformer | | FEDformer | | Autoformer | | Informer | |
|---|---|---|---|---|---|---|---|---|---|---|---|
| Metric | | MSE | MAE | MSE | MAE | MSE | MAE | MSE | MAE | MSE | MAE |
| ETTh | 12 | **0.2826** | **0.3383** | 0.4479 | 0.4582 | 0.3272 | 0.3940 | 0.5016 | 0.5204 | 0.4299 | 0.4644 |
| | 24 | **0.3430** | **0.3816** | 0.4602 | 0.4621 | 0.3699 | 0.4185 | 0.5063 | 0.5309 | 0.4880 | 0.4963 |
| | 48 | **0.3705** | **0.3939** | 0.4855 | 0.4735 | 0.3912 | 0.4347 | 0.5703 | 0.5563 | 0.6625 | 0.5774 |
| | 96 | **0.4039** | **0.4074** | 0.5090 | 0.4851 | 0.4194 | 0.4476 | 0.6052 | 0.5663 | 0.9584 | 0.7157 |
| | 128 | **0.4270** | **0.4208** | 0.5279 | 0.4949 | 0.4360 | 0.4551 | 0.6043 | 0.5726 | 0.9504 | 0.7197 |
| | 168 | **0.4455** | **0.4363** | 0.5446 | 0.5044 | 0.4733 | 0.4783 | 0.7382 | 0.6199 | 1.1043 | 0.7867 |
| Weather | 12 | **0.0811** | **0.1199** | 0.0900 | 0.1537 | 0.1476 | 0.2350 | 0.2042 | 0.2960 | 0.2351 | 0.3128 |
| | 24 | **0.1065** | **0.1484** | 0.1396 | 0.2224 | 0.1624 | 0.2496 | 0.2200 | 0.3141 | 0.1244 | 0.2022 |
| | 48 | **0.1290** | **0.1784** | 0.1848 | 0.2735 | 0.1993 | 0.2898 | 0.2691 | 0.3542 | 0.2352 | 0.3129 |
| | 96 | **0.1633** | **0.2151** | 0.2034 | 0.2994 | 0.2350 | 0.3139 | 0.2891 | 0.3673 | 0.2808 | 0.3586 |
| | 128 | **0.1774** | **0.2283** | 0.2092 | 0.2972 | 0.2395 | 0.3148 | 0.2758 | 0.3469 | 0.3055 | 0.3723 |
| | 168 | **0.2031** | **0.2525** | 0.2199 | 0.3016 | 0.2632 | 0.3281 | 0.2861 | 0.3506 | 0.3473 | 0.4003 |
| Exchange | 24 | 0.0276 | 0.1167 | **0.0266** | **0.1130** | 0.0717 | 0.1958 | 0.0894 | 0.2239 | 0.4963 | 0.5623 |
| | 48 | **0.0438** | 0.1481 | 0.0441 | **0.1464** | 0.0954 | 0.2247 | 0.1474 | 0.2881 | 1.0477 | 0.8169 |
| | 96 | **0.0814** | 0.2074 | 0.0861 | **0.2044** | 0.1470 | 0.2790 | 0.2883 | 0.3957 | 1.1038 | 0.8215 |
| | 128 | **0.1108** | **0.2361** | 0.1153 | 0.2373 | 0.1886 | 0.3153 | 0.3102 | 0.4107 | 1.1978 | 0.8535 |
| | 168 | **0.1443** | 0.2824 | 0.1549 | **0.2773** | 0.2484 | 0.3638 | 0.3066 | 0.4108 | 1.1564 | 0.8444 |
| | 196 | **0.1661** | 0.3040 | 0.1830 | **0.3034** | 0.2718 | 0.3800 | 0.2990 | 0.4021 | 1.1679 | 0.8545 |
| ILI | 24 | **2.4781** | **0.9925** | 3.1358 | 1.2128 | 3.3017 | 1.2689 | 3.3292 | 1.2088 | 4.2526 | 1.3551 |
| | 36 | **2.2103** | **0.8956** | 2.9369 | 1.1218 | 2.6125 | 1.0575 | 3.4076 | 1.1688 | 4.7647 | 1.4433 |
| | 48 | **1.9697** | **0.8826** | 2.9386 | 1.1120 | 2.5883 | 1.0683 | 3.2077 | 1.1125 | 4.8189 | 1.4553 |
| | 60 | **2.3496** | **0.9545** | 2.8840 | 1.1324 | 2.8460 | 1.1533 | 3.3373 | 1.1659 | 4.7974 | 1.4669 |
| | 72 | **2.1563** | **0.8884** | 2.8615 | 1.1579 | 2.8921 | 1.1721 | 3.1079 | 1.1237 | 4.1188 | 1.3718 |
| | 96 | **2.3860** | **0.9827** | 3.1109 | 1.2186 | 3.1048 | 1.2412 | 3.0530 | 1.1260 | 4.5218 | 1.4401 |

**Sampling Time** Tables 3b and 3c study the influence of sampling time in each iteration and data augmentation on Ti-MAE training stage. Ti-MAE works well with proper sampling time in each iteration and even no extra data augmentation, which is different from other existing representation learning methods on time series, especially contrastive learning models which rely on heavily data augmentation. Ti-MAE can directly learn adequate information from masked data. Additionally, introducing extra data augmentation will degrade the performance due to inevitable distortions of the original data, which is different from the result of MAE for images or videos. Random masking in each iteration generates a large number of different views without any distortion so that model can make use of visible tokens to capture more useful features.

Table 3: Ablation experiments on Weather. The entries marked in **bold** are the same which specify the default settings. Lower MSE and MAE represent better performance. This table format follows (Feichtenhofer et al., 2022).

(a) Masking ratio

| Ratio | MSE | MAE |
|---|---|---|
| 0.45 | 0.3082 | 0.3557 |
| 0.60 | 0.2650 | 0.3414 |
| **0.75** | **0.2103** | **0.2696** |
| 0.90 | 0.2483 | 0.3176 |

(b) Sampling time

| #Sampling | MSE | MAE |
|---|---|---|
| 20 | 0.2151 | 0.2892 |
| 25 | 0.2447 | 0.3095 |
| **30** | **0.2103** | **0.2696** |
| 35 | 0.2171 | 0.2769 |

(c) Data augmentation

| Case | MSE | MAE |
|---|---|---|
| **None** | **0.2103** | **0.2696** |
| Scaling | 0.2383 | 0.3022 |
| Shifting | 0.2399 | 0.3388 |
| Jittering | 0.2508 | 0.3316 |

(d) Input sequence length

| Length | MSE | MAE |
|---|---|---|
| 200 | 0.2413 | 0.2844 |
| **300** | **0.2103** | **0.2696** |
| 400 | 0.2877 | 0.3501 |
| 500 | 0.2328 | 0.2985 |

(e) Decoder width

| #Blocks | MSE | MAE |
|---|---|---|
| 1 | 0.2375 | 0.2954 |
| **2** | **0.2103** | **0.2696** |
| 3 | 0.2125 | 0.3112 |
| 4 | 0.2616 | 0.3057 |

(f) Decoder depth

| Dim | MSE | MAE |
|---|---|---|
| 16 | 0.4374 | 0.4805 |
| **32** | **0.2103** | **0.2696** |
| 64 | 0.2380 | 0.2813 |
| 128 | 0.3172 | 0.3758 |

**Input sequence length.** In Table 3d we compare different length of input time series in training stage. Surprisingly, although lengthening the input length of the pre-training stage can improve the performance within limits, too long input sequence may degrade the results of our model because there is a certain conflict between the complex periodic pattern in the long sequence and the short-term prediction task in the downstream.

**Decoder Design.** Tables 3e and 3f show the influence of the decoder width and depth. A shallow design of the decoder is sufficient for reconstruction tasks. It is because that time series data are not that complicated and thus need lower decoding dimension to reduce redundancy. Such a lightweight decoder can efficiently reduce computational complexity and memory usage.

## 4.3 TIME SERIES CLASSIFICATION

In the previous section, we have improved the performance of our framework on forecasting tasks by reducing the consistency between upstream tasks and downstream tasks compared to contrastive learning methods. Thus, we should evaluate learning ability of instance-level representation on classification tasks. The results on 128 UCR archive are summarized in Table 4. Compared to other representation learning methods, Ti-MAE achieves more compatible performance. More details and full results of each dataset in UCR are listed in the appendix. Following (Yue et al., 2022), Critical Difference diagram (Demsar, 2006) for Nemenyi tests conducted on all datasets is shown as Figure 6, where classifiers that are connected by a bold line do not have a significant difference. This proves that Ti-MAE could learn good instance-level representations directly from the raw time series data without any hierarchical tricks or data augmentation.

Table 4: 128 UCR Archive Classification

| method | Ti-MAE | TS2Vec | T-Loss | TS-TCC | TST | TNC | DTW |
|---|---|---|---|---|---|---|---|
| Avg.Acc. | **0.8231** | 0.8201 | 0.7875 | 0.7396 | 0.6385 | 0.7431 | 0.7278 |
| Avg.Rank | **2.054** | 3.016 | 4.016 | 4.445 | 4.883 | 3.875 | 5.711 |

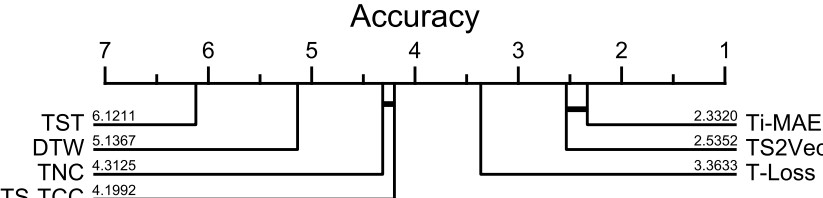

Figure 6: Critical Difference (CD) diagram on UCR classification with a 95% confidence level.

## 5 CONCLUSION

This paper proposes a novel self-supervised framework named Ti-MAE for time series representation learning, which randomly masks out tokenized time series and learns an autoencoder to reconstruct them at the point-level. Ti-MAE bridges the connection between contrastive representation learning and generative Transformer-based methods and greatly improves the performance on forecasting tasks due to reducing the inconsistency of upstream and downstream tasks compared to contrastive learning methods. Compared with the fixed continuous masking strategy used in existing Transformer-based models, Ti-MAE adequately leverages all the input sequence and alleviates the distribution shift problem. The flexible setting of masking ratio makes Ti-MAE more adaptive to various prediction scenarios with different time steps. The experiments on real-world datasets and ablation study demonstrate the effectiveness and scalability of our framework. Future work will extend our work for different reconstruction targets according to their requirements.

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

# A EXPERIMENTAL DETAILS

## A.1 REPRODUCTION DETAILS FOR TI-MAE

The default settings of Ti-MAE are shown in Table 5 in detail. We use one Conv1d layer with the setting of $\text{kernel} = 3, \text{stride} = 1, \text{padding} = 1$ to obtain the encoder input embedding, and then we add a fixed positional encoding as

$$
\begin{aligned}
\text{PE}(pos, 2i) &= \sin(\frac{pos}{10000^{2i/\text{d}_{\text{model}}}}) \\
\text{PE}(pos, 2i+1) &= \cos(\frac{pos}{10000^{2i/\text{d}_{\text{model}}}}),
\end{aligned}
\tag{2}
$$

where $\text{d}_{\text{model}}$ represents the number of hidden states. After encoder input embedding, we randomly mask out 75% tokens, and then remaining visible parts are fed into the encoder. The encoder and decoder of Ti-MAE both contain 2 Transformer blocks as widely adopted in Devlin et al. (2019); Dosovitskiy et al. (2021), each of which consists of one vanilla self-attention layer with 4 heads and a point-wise feed forward layer. As recommended in Dosovitskiy et al. (2021), we adopt pre-norm instead of post-norm for stability of the model in training stage. Equation 3 demonstrates the whole process in the encoder:

$$
\begin{aligned}
\mathbf{Z}_i^d &= \text{RandomMask}(\text{Conv1d}(\mathcal{X}_{l,n}) + \text{PE}(\mathcal{X}_{l,n})) \\
\hat{\mathbf{Z}}_i^d &= \mathbf{Z}_i^d + \text{MHSA}(\text{LayerNorm}(\mathbf{Z}_i^d, \mathbf{Z}_i^d, \mathbf{Z}_i^d)) \\
\tilde{\mathbf{Z}}_i^d &= \hat{\mathbf{Z}}_i^d + \text{MLP}(\text{LayerNorm}(\hat{\mathbf{Z}}_i^d))
\end{aligned}
\tag{3}
$$

where we use $\mathcal{X}_{l,n}$ to denote the vectors in dimension $n$ with the length of $l$, and $Z_i^d$ to denote the intermediate representation in dimension $d$ with the length of $i$. In the decoder, we first apply a linear layer to reduce the input dimension to $d'$ ($64 \to 32$) for training efficiency. Given the position to be reconstruct, zero initialized masked tokens are padded to the encoded tokens with the original positional encoding. A dropout layer ($p = 0.1$) is added to the bottom of Transformer blocks to prevent the over-fitting problem. The last linear projection layer of the decoder is to reconstruct the missing values at the point-level. Equation 4 demonstrates the whole process of the decoder:

$$
\begin{aligned}
\mathbf{Z}_l^{d'} &= \text{Padding}(\text{Linear}(\tilde{\mathbf{Z}}_i^d)) + \text{PE}(\mathcal{X}_{l,d'})) \\
\hat{\mathbf{Z}}_l^{d'} &= \mathbf{Z}_l^{d'} + \text{MHSA}(\text{LayerNorm}(\mathbf{Z}_l^{d'}, \mathbf{Z}_l^{d'}, \mathbf{Z}_l^{d'})) \\
\tilde{\mathbf{Z}}_l^{d'} &= \hat{\mathbf{Z}}_l^{d'} + \text{MLP}(\text{LayerNorm}(\hat{\mathbf{Z}}_l^{d'})) \\
\tilde{\mathcal{X}}_{l,n} &= \text{Projection}(\tilde{\mathbf{Z}}_l^{d'})
\end{aligned}
\tag{4}
$$

where we use $Z_l^{d'}$ to denote the intermediate representation in dimension $d'$ with the length of $l$, and $\tilde{\mathcal{X}}_{l,n}$ to denote our reconstruction goals.

Table 5: Default settings of Ti-MAE

| Config | Value |
|---|---|
| optimizer | Adam Kingma & Ba (2015) |
| learning rate | 0.001 |
| learning rate schedule | cosine decay |
| epochs | 10 |
| masking ratio | 75% |
| sampling time | 30 |
| batch size | 64 |
| #encoder layer | 2 |
| #decoder layer | 2 |
| $\text{d}_{\text{model}}$ | 64 |
| dropout | 0.1 |

Notably, all the linear layers in Ti-MAE are initialized through xavier Glorot & Bengio (2010). The choice of Transformer blocks in Ti-MAE is flexible so that you can use other designed blocks with more inductive biases if necessary.

## A.2 DETAILS ON BASELINES

For forecasting tasks, the results of CoST Woo et al. (2022a), TS2Vec Yue et al. (2022), TNC Tonekaboni et al. (2021), MoCo Chen et al. (2021), Autoformer Wu et al. (2021), Informer Zhou et al. (2021), ETSformer Woo et al. (2022b) and FEDformer Zhou et al. (2022) are all based on our reproduction. For classification tasks, the results of TS2Vec are based on our reproduction. Other results of classification are directly taken from Yue et al. (2022).

CoST Woo et al. (2022a) was recently proposed as a contrastive learning framework of disentangled seasonal-trend representations for time series forecasting. They comprises both time domain and frequency domain contrastive losses to learn discriminative trend and seasonal representations. We use the public official source code from `https://github.com/salesforce/CoST`.

TS2Vec Yue et al. (2022) is a universal framework for learning representations of time series in an arbitrary semantic level through applying contrastive learning in a hierarchical way over augmented context views. TS2Vec can obtain timestamp-level and instance-level representations for forecasting and classification simultaneously. We take the officially implemented code from `https://github.com/yuezhihan/ts2vec`.

TNC Tonekaboni et al. (2021) is a self-supervised contrastive learning framework for time series, where the positive samples come from the neighboring similar signals. We use the official open source code from `https://github.com/sanatonek/TNCrepresentationlearning` and all the settings of hyper-parameters follows Woo et al. (2022a).

MoCo Chen et al. (2021) is a self-supervised contrastive learning framework widely used in computer vision domain, which uses a dynamic queue to save a large number of positive and negative samples with consistency. We directly apply this framework on time series data using the official code from `https://github.com/facebookresearch/moco`. Hyper-parameters are the same as Woo et al. (2022a).

Autoformer Wu et al. (2021) is a novel end-to-end supervised model with a decomposition architecture for time series forecasting. By directly subtracting trend parts obtained from moving average, they design an auto-correlation mechanism as a replacement for self-attention to capture long-term dependencies from seasonality parts. We use their open source code from `https://github.com/thuml/Autoformer`. Hyper-parameters are remain the default values in the code.

Informer Zhou et al. (2021) is an efficient end-to-end supervised model for time series forecasting. They propose a novel sparse attention to reduce time complexity and memory usage. We take the officially implemented code from `https://github.com/zhouhaoyi/Informer2020`. Hyper-parameters are used as suggested in their paper.

ETSformer Woo et al. (2022b) proposes an interpretable Transformer architecture which decomposes forecasts into level, growth, and seasonality components. And they employ both exponential smoothing attention and frequency attention to reduce computational complexity. We use their open source code from `https://github.com/salesforce/ETSformer`. Hyper-parameters are used as suggested in their paper.

FEDformer Zhou et al. (2022) proposes to combine Transformer with the seasonal-trend decomposition method, and exploit the fact that most time series tend to have a sparse representation in well-known basis such as Fourier transform, and develop a frequency enhanced Transformer. We use the official code from `https://github.com/MAZiqing/FEDformer`. Hyper-parameters are remain the default values in the code.

## A.3 DETAILS ON BENCHMARK TASKS

For time series forecasting tasks, the evaluation settings of end-to-end supervised models and other representation learning methods are slightly different. For other representation learning methods, we follow Yue et al. (2022) to evaluate the performance of their models. Specifically, we use a ridge regression trained on the learned representations to predict the future values. The regularization term $\alpha$ is selected by grid search from $\{0.1, 0.2, 0.5, 1, 2, 5, 10, 20, 50, 100, 200, 500, 1000\}$. It is important to stress that Ti-MAE can directly generate future values from its decoder without any

extra regressor (e.g. setting 50% masking ratio means giving one half of the entries to predict the other half.). As for end-to-end models, we set the length of input sequence as 96 to predict future time series with different horizons. Notably, for fair comparison with other SOTA Transformer-based methods including FEDformer and ETSformer on forecasting, we have fine-tuned Ti-MAE on forecasting tasks. Specifically, we extract the encoder of Ti-MAE and freeze it after pre-training, and add an extra linear regressor for fine-tuning.

For classification tasks, we directly obtain instance-level representations by average or max pooling over all timestamps following Yue et al. (2022). To evaluate the performance of models on classification, we follow the same protocol Franceschi et al. (2019), where an SVM classifier with RBF kernel is trained on obtained instance-level representations. The full results of each dataset in UCR are provided in Table 13 and 14.

Notably, due to the flexible design of the Transformer block, we can utilize any layer of the encoder or the decoder of Ti-MAE to get intermediate representations. Extra class token is also a choice if necessary. In our experiments, we simply gather the encoder embedding of Ti-MAE as instance-level representations for evaluation. To accelerate the training of the model, we perform equidistant sampling for different datasets to reduce input to less than 1024 for training efficiency.

Table 6: The classification results on morphological datasets with or without positional encoding

| dataset | w/ PE | w/o PE |
|---|---|---|
| OSULeaf | 0.59 | **0.74** |
| ShapeletSim | 0.54 | **0.91** |
| Worms | 0.64 | **0.78** |

TS2Vec also reports an interesting phenomenon that using Transformer instead of Dilated CNN as backbone will largely degrade the performance on classification tasks. We also find similar problems, especially on morphological datasets. We suppose that some morphological datasets have almost no seasonality, while the local morphological characteristics determine the data classification. The positional encoding introduced in the encoder may destroy these morphological features. Simply removing position embedding in the encoder when generating representations will significantly affect the performance of classification. Table 6 shows the classification results on some morphological datasets with or without position embedding.

# B ADDITIONAL EXPERIMENTAL RESULTS

## B.1 THE IMPACT OF MASKING RATIO AND SAMPLING STRATEGIES

Table 7: The impact of masking ratio on forecasting tasks.

| Method | | ETTh | | Weather | | Excahnge | | ILI | |
|---|---|---|---|---|---|---|---|---|---|
| Metric | | MSE | MAE | MSE | MAE | MSE | MAE | MSE | MAE |
| Masking | 30% | 0.6181 | 0.4984 | 0.4320 | 0.4698 | 0.2707 | 0.3531 | 2.2039 | 0.9908 |
| | 45% | 0.5140 | 0.4830 | 0.3082 | 0.3557 | 0.2328 | 0.3380 | 2.0452 | 0.9843 |
| | 60% | 0.5011 | 0.4490 | 0.2650 | 0.3414 | 0.2239 | 0.3340 | 2.0389 | 0.9707 |
| | 75% | **0.4403** | **0.4338** | **0.2103** | **0.2696** | **0.1701** | **0.2972** | **2.0150** | 0.9646 |
| | 90% | 0.4597 | 0.4385 | 0.2483 | 0.3176 | 0.1952 | 0.3172 | 2.0332 | **0.9607** |

Table 8: The impact of different masking strategies with 75% ratio on Weather.

| Masking Strategy | Random | Continuous | Split | Periodic |
|---|---|---|---|---|
| MSE | **0.2103** | 0.3834 | 0.3564 | 0.2720 |
| MAE | **0.2696** | 0.4420 | 0.3936 | 0.3357 |

Table 7 summarizes the impact of masking ratio on different forecasting tasks under the setting of 200-100. We can see that the best masking ratio is generally around 75% given the continuous nature of time series data. Table 8 studies the impact of different masking strategies with 75% ratio on Weather dataset of 96-96 setting. Specifically, random masking means tokens are randomly masked; continuous masking means we only mask historical time series and reconstruct future values, which is the same as traditional forecasting methods; split masking means we both mask historical time series to reconstruct future values, and mask future time series to reconstruct historical sequence; periodic masking means tokens are periodically masked. Notably, periodic masked tokens with a length of four are sampled equidistantly to maintain the same masking ratio. We can see that random masking achieves the best result because randomness can adequately exploit the whole time series data with less inductive bias.

## B.2 RUNNING TIME ANALYSIS

Table 9 shows the running time in seconds for each stage of different Transformer-based methods, where we execute three times for each setting (using 96 historical steps to predict future steps of 24, 48, 96, 288 and 672 respectively). All experiments are performed on one single Nvidia V100 GPU. Although many Transformer-based models have $O(L \log L)$ complexity, however, there exists a large constant since these methods generally need to do a bulk of pre-treatment (e.g. Fourier Transform, Wavelet Transform), which makes the overall training not that efficient. In comparison, although our proposed Ti-MAE has $O(L^2)$ complexity due to the vanilla attention mechanism, we need to pre-train the encoder of Ti-MAE **only once** and can fine-tune it on different forecasting settings. Thus, the total running time of Ti-MAE is less than other Transformer-based methods.

Table 9: Running time (seconds) for Transformer-based methods at different stages.

| Stage | H | Ti-MAE | FEDformer | ETSformer | Autoformer | Informer |
|---|---|---|---|---|---|---|
| Pre-training | / | $335.1 \pm 0.5$ | / | / | / | / |
| Training | 24 | $17.5 \pm 0.2$ | $170.1 \pm 2.8$ | $68.6 \pm 0.7$ | $61.8 \pm 0.4$ | $68.4 \pm 0.7$ |
| | 48 | $17.6 \pm 0.1$ | $199.2 \pm 2.1$ | $69.1 \pm 0.6$ | $68.7 \pm 0.7$ | $76.1 \pm 0.6$ |
| | 96 | $17.9 \pm 0.2$ | $250.1 \pm 2.0$ | $72.1 \pm 0.5$ | $79.8 \pm 0.4$ | $87.7 \pm 0.2$ |
| | 288 | $18.6 \pm 0.2$ | $324.0 \pm 1.8$ | $73.1 \pm 1.0$ | $130.5 \pm 0.9$ | $137.0 \pm 0.3$ |
| | 672 | $19.8 \pm 0.3$ | $460.9 \pm 2.1$ | $76.3 \pm 0.6$ | $220.5 \pm 0.4$ | $220.1 \pm 0.2$ |
| Inference | 24 | $5.7 \pm 0.1$ | $9.1 \pm 0.4$ | $9.0 \pm 0.4$ | $13.6 \pm 0.2$ | $9.3 \pm 0.1$ |
| | 48 | $6.1 \pm 0.2$ | $10.9 \pm 0.4$ | $9.1 \pm 0.3$ | $15.1 \pm 0.2$ | $10.2 \pm 0.2$ |
| | 96 | $6.5 \pm 0.1$ | $12.4 \pm 0.1$ | $9.9 \pm 0.3$ | $18.0 \pm 0.3$ | $12.1 \pm 0.5$ |
| | 288 | $8.7 \pm 0.7$ | $16.7 \pm 0.4$ | $13.3 \pm 0.5$ | $31.3 \pm 0.5$ | $18.8 \pm 0.3$ |
| | 672 | $12.9 \pm 1.1$ | $25.2 \pm 1.3$ | $19.1 \pm 1.4$ | $55.6 \pm 0.9$ | $31.4 \pm 1.2$ |

## B.3 ABLATION STUDY ON TI-MAE'S COMPONENTS

Table 10 shows the impact of different components of Ti-MAE, which proves the effectiveness of random masking strategy, Transformer-based backbone and other necessary parts.

Table 10: Ablation study of Ti-MAE's components on the Exchange dataset (200-100 setting).

| Ablation variant | MSE | MAE |
|---|---|---|
| Default | **0.2111** | **0.3367** |
| Random $\rightarrow$ fixed continuous masking | 0.2505 (-18.7%) | 0.3700 (-9.9%) |
| Encoder w/o positional encoding | 0.3082 (-46.0%) | 0.3996 (-18.7%) |
| Decoder w/o positional encoding | 0.2276 (-7.8%) | 0.3518 (-4.5%) |
| pre-norm $\rightarrow$ post-norm | 0.2213 (-4.8%) | 0.3474 (-3.2%) |
| Transformer $\rightarrow$ TCN | 0.2340 (-10.8%) | 0.3558 (-5.7%) |
| Transformer $\rightarrow$ LSTM | 0.2406 (-14.0%) | 0.3612 (-7.3%) |

### B.4 FORECASTING RESULTS WITH DIFFERENT DIMENSION

The input time series and the output one do not need to have the same dimensionality. Actually the final linear projection layer in the decoder can easily project the input dimensionality to the desired out dimensionality. Table 11 shows the results of using multivariate time series to predict the last univariate target.

Table 11: Forecasting results with different dimension compared to representation learning methods.

| Method | | Ti-MAE | | CoST | | TS2Vec | | TNC | | MoCo | |
|---|---|---|---|---|---|---|---|---|---|---|---|
| Metric | | MSE | MAE | MSE | MAE | MSE | MAE | MSE | MAE | MSE | MAE |
| ETTh | 24 | **0.031** | **0.134** | 0.040 | 0.152 | 0.039 | 0.151 | 0.057 | 0.184 | 0.040 | 0.151 |
| | 48 | **0.048** | **0.167** | 0.060 | 0.186 | 0.062 | 0.189 | 0.094 | 0.239 | 0.063 | 0.191 |
| | 168 | **0.076** | **0.207** | 0.097 | 0.236 | 0.142 | 0.291 | 0.171 | 0.329 | 0.122 | 0.268 |
| | 336 | **0.097** | **0.240** | 0.112 | 0.306 | 0.160 | 0.316 | 0.192 | 0.357 | 0.144 | 0.297 |
| Weather | 24 | **0.005** | **0.056** | 0.096 | 0.213 | 0.096 | 0.215 | 0.102 | 0.221 | 0.097 | 0.216 |
| | 48 | **0.012** | **0.087** | 0.138 | 0.262 | 0.140 | 0.264 | 0.139 | 0.264 | 0.140 | 0.264 |
| | 168 | **0.014** | **0.108** | 0.207 | 0.334 | 0.207 | 0.335 | 0.198 | 0.328 | 0.198 | 0.326 |
| | 336 | **0.015** | **0.115** | 0.230 | 0.356 | 0.231 | 0.360 | 0.215 | 0.347 | 0.220 | 0.350 |

### B.5 TRANSFERABILITY STUDY

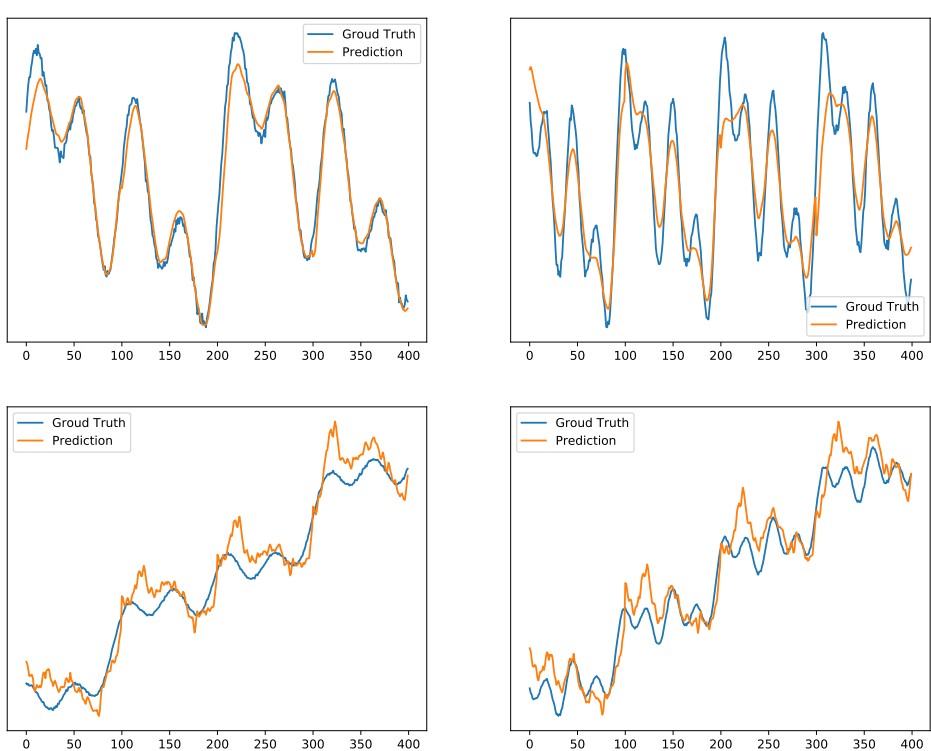

Figure 7: Transferability of Ti-MAE on different trend and seasonality patterns.

To evaluate the transferability of our framework, we generate a set of time series data with different trend and seasonality patterns, which follows

$$y(t) = \cos(\alpha \cdot t) + \cos(\frac{\alpha}{2} \cdot t) + \cos(\frac{\alpha}{4} \cdot t) + \beta \cdot t + \epsilon \tag{5}$$

where the hyper-parameters $\alpha$ and $\beta$ respectively control the trend and seasonality patterns, and the noises $\epsilon \sim N(0, 0.1)$. We train our Ti-MAE under the setting of $\alpha = 300, \beta = 3$ and evaluate the forecasting performance on other different settings. Table 8 and Figure 7 demonstrate the strong transferability of Ti-MAE under different trend and seasonality patterns.

Table 12: The results of forecasting 400 time steps on simulated time series data with different trend and seasonality patterns.

| Setting | $\alpha = 300$ $\beta = 3$ | $\alpha = 600$ $\beta = 3$ | $\alpha = 300$ $\beta = 100$ | $\alpha = 600$ $\beta = 100$ |
|---------|--------|--------|--------|--------|
| MSE | 0.0134 | 0.0596 | 0.0089 | 0.0232 |
| MAE | 0.0778 | 0.1912 | 0.0881 | 0.0711 |

Table 13: Full classification results on 128 UCR datasets part 1.

| Dataset | TMAE | TS2Vec | T-Loss | TNC | TS-TCC | TST | DTW |
|---|---|---|---|---|---|---|---|
| ACSF1 | 0.820 | 0.870 | **0.900** | 0.730 | 0.730 | 0.760 | 0.640 |
| Adiac | **0.788** | 0.726 | 0.675 | 0.726 | 0.767 | 0.550 | 0.604 |
| AllGestureWiimoteX | 0.633 | **0.782** | 0.763 | 0.703 | 0.697 | 0.259 | 0.716 |
| AllGestureWiimoteY | 0.682 | **0.791** | 0.726 | 0.699 | 0.741 | 0.423 | 0.729 |
| AllGestureWiimoteZ | 0.671 | **0.760** | 0.723 | 0.646 | 0.689 | 0.447 | 0.643 |
| ArrowHead | **0.874** | 0.794 | 0.766 | 0.703 | 0.737 | 0.771 | 0.703 |
| BME | **1.000** | 0.987 | 0.993 | 0.973 | 0.933 | 0.760 | 0.900 |
| Beef | **0.900** | 0.700 | 0.667 | 0.733 | 0.600 | 0.500 | 0.633 |
| BeetleFly | 0.900 | 0.750 | 0.800 | 0.850 | 0.800 | **1.000** | 0.700 |
| BirdChicken | **1.000** | 0.800 | 0.850 | 0.750 | 0.650 | 0.650 | 0.750 |
| CBF | **1.000** | **1.000** | 0.983 | 0.983 | 0.998 | 0.898 | 0.997 |
| Car | **0.867** | 0.800 | 0.833 | 0.683 | 0.583 | 0.55 | 0.733 |
| Chinatown | **0.985** | 0.974 | 0.951 | 0.977 | 0.983 | 0.936 | 0.957 |
| ChlorineConcentration | 0.725 | **0.804** | 0.749 | 0.760 | 0.753 | 0.562 | 0.648 |
| CinCECGTorso | **0.971** | 0.793 | 0.713 | 0.669 | 0.671 | 0.508 | 0.651 |
| Coffee | **1.000** | **1.000** | **1.000** | **1.000** | **1.000** | 0.821 | **1.000** |
| Computers | **0.780** | 0.648 | 0.664 | 0.684 | 0.704 | 0.696 | 0.700 |
| CricketX | 0.674 | **0.777** | 0.713 | 0.623 | 0.731 | 0.385 | 0.754 |
| CricketY | 0.659 | **0.769** | 0.728 | 0.597 | 0.718 | 0.467 | 0.744 |
| CricketZ | 0.718 | **0.810** | 0.708 | 0.682 | 0.713 | 0.403 | 0.754 |
| Crop | 0.751 | **0.756** | 0.722 | 0.738 | 0.742 | 0.710 | 0.665 |
| DiatomSizeReduction | 0.984 | 0.990 | 0.984 | **0.993** | 0.977 | 0.961 | 0.967 |
| DistalPhalanxOutlineAgeGroup | 0.763 | 0.719 | 0.727 | 0.741 | 0.755 | 0.741 | **0.770** |
| DistalPhalanxOutlineCorrect | **0.793** | 0.754 | 0.775 | 0.754 | 0.754 | 0.728 | 0.717 |
| DistalPhalanxTW | **0.727** | 0.698 | 0.676 | 0.669 | 0.676 | 0.568 | 0.590 |
| DodgerLoopDay | **0.613** | 0.538 | 0.241 | 0.183 | 0.206 | 0.200 | 0.500 |
| DodgerLoopGame | 0.739 | **0.826** | 0.415 | 0.508 | 0.493 | 0.696 | 0.877 |
| DodgerLoopWeekend | **0.978** | 0.949 | 0.623 | 0.684 | 0.601 | 0.732 | 0.949 |
| ECG200 | 0.910 | 0.860 | **0.940** | 0.830 | 0.880 | 0.830 | 0.770 |
| ECG5000 | **0.942** | 0.932 | 0.933 | 0.937 | 0.941 | 0.928 | 0.924 |
| ECGFiveDays | 0.988 | **1.000** | **1.000** | 0.999 | 0.878 | 0.763 | 0.768 |
| EOGHorizontalSignal | 0.558 | 0.528 | **0.605** | 0.442 | 0.401 | 0.373 | 0.503 |
| EOGVerticalSignal | **0.547** | 0.483 | 0.434 | 0.392 | 0.376 | 0.298 | 0.448 |
| Earthquakes | **0.748** | **0.748** | **0.748** | **0.748** | **0.748** | **0.748** | 0.719 |
| ElectricDevices | 0.685 | **0.724** | 0.707 | 0.700 | 0.686 | 0.676 | 0.602 |
| EthanolLevel | **0.744** | 0.388 | 0.382 | 0.424 | 0.486 | 0.260 | 0.276 |
| FaceAll | **0.880** | 0.789 | 0.786 | 0.766 | 0.813 | 0.504 | 0.808 |
| FaceFour | 0.875 | 0.852 | **0.920** | 0.659 | 0.773 | 0.511 | 0.830 |
| FacesUCR | 0.866 | **0.929** | 0.884 | 0.789 | 0.863 | 0.543 | 0.905 |
| FiftyWords | **0.787** | 0.754 | 0.732 | 0.653 | 0.653 | 0.525 | 0.690 |
| Fish | 0.897 | **0.920** | 0.891 | 0.817 | 0.817 | 0.720 | 0.823 |
| FordA | 0.818 | **0.940** | 0.928 | 0.902 | 0.930 | 0.568 | 0.555 |
| FordB | 0.652 | 0.802 | 0.793 | 0.733 | **0.815** | 0.507 | 0.620 |
| FreezerRegularTrain | 0.987 | 0.984 | 0.956 | **0.991** | 0.989 | 0.922 | 0.899 |
| FreezerSmallTrain | 0.959 | 0.872 | 0.933 | 0.982 | **0.979** | 0.920 | 0.753 |
| Fungi | 0.968 | 0.935 | **1.000** | 0.527 | 0.753 | 0.366 | 0.839 |
| GestureMidAirD1 | **0.662** | 0.592 | 0.608 | 0.431 | 0.369 | 0.208 | 0.569 |
| GestureMidAirD2 | **0.546** | 0.523 | **0.546** | 0.362 | 0.254 | 0.138 | 0.608 |
| GestureMidAirD3 | **0.400** | 0.323 | 0.285 | 0.292 | 0.177 | 0.154 | 0.323 |
| GesturePebbleZ1 | **0.901** | 0.849 | 0.919 | 0.378 | 0.395 | 0.500 | 0.791 |
| GesturePebbleZ2 | **0.918** | 0.854 | 0.899 | 0.316 | 0.430 | 0.380 | 0.671 |
| GunPoint | **0.993** | 0.973 | 0.980 | 0.967 | **0.993** | 0.827 | 0.907 |
| GunPointAgeSpan | **0.994** | 0.962 | **0.994** | 0.984 | **0.994** | 0.991 | 0.918 |
| GunPointMaleVersusFemale | 0.997 | **1.000** | 0.997 | 0.994 | 0.997 | **1.000** | 0.997 |
| GunPointOldVersusYoung | **1.000** | **1.000** | **1.000** | **1.000** | **1.000** | **1.000** | 0.838 |
| Ham | **0.800** | 0.714 | 0.724 | 0.752 | 0.743 | 0.524 | 0.467 |
| HandOutlines | 0.919 | 0.919 | 0.922 | **0.930** | 0.724 | 0.735 | 0.881 |
| Haptics | 0.484 | **0.519** | 0.490 | 0.474 | 0.396 | 0.357 | 0.377 |
| Herring | **0.656** | 0.609 | 0.594 | 0.594 | 0.594 | 0.594 | 0.531 |
| HouseTwenty | **0.941** | 0.899 | 0.933 | 0.782 | 0.790 | 0.815 | 0.924 |
| InlineSkate | 0.380 | **0.403** | 0.371 | 0.378 | 0.347 | 0.287 | 0.384 |
| InsectEPGRegularTrain | **1.000** | **1.000** | **1.000** | **1.000** | **1.000** | **1.000** | 0.872 |
| InsectEPGSmallTrain | **1.000** | **1.000** | **1.000** | **1.000** | **1.000** | **1.000** | 0.735 |
| InsectWingbeatSound | **0.639** | 0.616 | 0.597 | 0.549 | 0.415 | 0.266 | 0.355 |

Table 14: Full classification results on 128 UCR datasets part 2.

| Dataset | TMAE | TS2Vec | T-Loss | TNC | TS-TCC | TST | DTW |
|---|---|---|---|---|---|---|---|
| ItalyPowerDemand | **0.967** | 0.932 | 0.954 | 0.928 | 0.955 | 0.845 | 0.950 |
| LargeKitchenAppliances | 0.787 | **0.869** | 0.789 | 0.776 | 0.848 | 0.595 | 0.795 |
| Lightning2 | 0.836 | **0.869** | **0.869** | **0.869** | 0.836 | 0.705 | **0.869** |
| Lightning7 | **0.808** | 0.781 | 0.795 | 0.767 | 0.685 | 0.411 | 0.726 |
| Mallat | **0.956** | 0.904 | 0.951 | 0.871 | 0.922 | 0.713 | 0.934 |
| Meat | **0.967** | **0.967** | 0.950 | 0.917 | 0.883 | 0.900 | 0.933 |
| MedicalImages | 0.771 | **0.799** | 0.750 | 0.754 | 0.747 | 0.632 | 0.737 |
| MelbournePedestrian | 0.949 | **0.958** | 0.944 | 0.942 | 0.949 | 0.741 | 0.791 |
| MiddlePhalanxOutlineAgeGroup | **0.675** | 0.643 | 0.656 | 0.643 | 0.630 | 0.617 | 0.500 |
| MiddlePhalanxOutlineCorrect | 0.811 | **0.831** | 0.825 | 0.818 | 0.818 | 0.753 | 0.698 |
| MiddlePhalanxTW | **0.623** | 0.578 | 0.591 | 0.571 | 0.610 | 0.506 | 0.506 |
| MixedShapesRegularTrain | **0.922** | 0.917 | 0.905 | 0.911 | 0.855 | 0.879 | 0.842 |
| MixedShapesSmallTrain | **0.875** | 0.854 | 0.860 | 0.813 | 0.735 | 0.828 | 0.780 |
| MoteStrain | **0.913** | 0.859 | 0.851 | 0.825 | 0.843 | 0.768 | 0.835 |
| NonInvasiveFetalECGThorax1 | 0.918 | **0.924** | 0.878 | 0.898 | 0.898 | 0.471 | 0.790 |
| NonInvasiveFetalECGThorax2 | 0.938 | **0.939** | 0.919 | 0.912 | 0.913 | 0.832 | 0.865 |
| OSULeaf | 0.736 | **0.851** | 0.760 | 0.723 | 0.723 | 0.545 | 0.591 |
| OliveOil | **0.933** | 0.867 | 0.867 | 0.833 | 0.800 | 0.800 | 0.833 |
| PLAID | 0.458 | 0.555 | 0.555 | 0.495 | 0.445 | 0.419 | **0.840** |
| PhalangesOutlinesCorrect | 0.772 | **0.806** | 0.784 | 0.787 | 0.804 | 0.773 | 0.728 |
| Phoneme | 0.229 | **0.296** | 0.276 | 0.180 | 0.242 | 0.139 | 0.228 |
| PickupGestureWiimoteZ | **0.840** | 0.800 | 0.740 | 0.620 | 0.600 | 0.240 | 0.660 |
| PigAirwayPressure | 0.240 | **0.807** | 0.510 | 0.413 | 0.380 | 0.120 | 0.106 |
| PigArtPressure | 0.760 | **0.966** | 0.928 | 0.808 | 0.524 | 0.774 | 0.245 |
| PigCVP | 0.750 | **0.813** | 0.788 | 0.649 | 0.615 | 0.596 | 0.154 |
| Plane | **1.000** | 0.990 | 0.990 | **1.000** | **1.000** | 0.933 | **1.000** |
| PowerCons | **1.000** | 0.967 | 0.900 | 0.933 | 0.961 | 0.911 | 0.878 |
| ProximalPhalanxOutlineAgeGroup | **0.863** | 0.834 | 0.844 | 0.854 | 0.839 | 0.854 | 0.805 |
| ProximalPhalanxOutlineCorrect | 0.876 | **0.890** | 0.859 | 0.866 | 0.873 | 0.770 | 0.784 |
| ProximalPhalanxTW | **0.829** | 0.790 | 0.771 | 0.810 | 0.800 | 0.780 | 0.761 |
| RefrigerationDevices | **0.611** | 0.603 | 0.515 | 0.565 | 0.563 | 0.483 | 0.464 |
| Rock | 0.660 | 0.660 | 0.580 | 0.580 | 0.600 | **0.680** | 0.600 |
| ScreenType | **0.579** | 0.411 | 0.416 | 0.509 | 0.419 | 0.419 | 0.397 |
| SemgHandGenderCh2 | 0.838 | **0.960** | 0.890 | 0.882 | 0.837 | 0.725 | 0.802 |
| SemgHandMovementCh2 | 0.700 | **0.862** | 0.789 | 0.593 | 0.613 | 0.420 | 0.584 |
| SemgHandSubjectCh2 | 0.813 | **0.947** | 0.853 | 0.771 | 0.753 | 0.484 | 0.727 |
| ShakeGestureWiimoteZ | 0.900 | **0.940** | 0.920 | 0.820 | 0.860 | 0.760 | 0.860 |
| ShapeletSim | 0.911 | **0.939** | 0.672 | 0.589 | 0.683 | 0.489 | 0.650 |
| ShapesAll | 0.840 | **0.890** | 0.848 | 0.788 | 0.773 | 0.733 | 0.768 |
| SmallKitchenAppliances | **0.741** | 0.733 | 0.677 | 0.725 | 0.691 | 0.592 | 0.643 |
| SmoothSubspace | **0.993** | 0.980 | 0.960 | 0.913 | 0.953 | 0.827 | 0.827 |
| SonyAIBORobotSurface1 | **0.912** | 0.910 | 0.902 | 0.804 | 0.899 | 0.724 | 0.725 |
| SonyAIBORobotSurface2 | **0.934** | 0.897 | 0.889 | 0.834 | 0.907 | 0.745 | 0.831 |
| StarLightCurves | **0.972** | 0.971 | 0.964 | 0.968 | 0.967 | 0.949 | 0.907 |
| Strawberry | **0.970** | 0.967 | 0.954 | 0.951 | 0.965 | 0.916 | 0.941 |
| SwedishLeaf | **0.938** | 0.923 | 0.914 | 0.880 | 0.923 | 0.738 | 0.792 |
| Symbols | 0.961 | **0.981** | 0.963 | 0.885 | 0.916 | 0.786 | 0.950 |
| SyntheticControl | 0.993 | 0.997 | 0.987 | **1.000** | 0.990 | 0.490 | 0.993 |
| ToeSegmentation1 | 0.890 | 0.925 | **0.939** | 0.864 | 0.930 | 0.807 | 0.772 |
| ToeSegmentation2 | **0.908** | 0.900 | 0.900 | 0.831 | 0.877 | 0.615 | 0.838 |
| Trace | **1.000** | **1.000** | 0.990 | **1.000** | **1.000** | **1.000** | **1.000** |
| TwoLeadECG | 0.985 | 0.982 | **0.999** | 0.993 | 0.976 | 0.871 | 0.905 |
| TwoPatterns | 0.994 | **1.000** | 0.999 | **1.000** | 0.999 | 0.466 | **1.000** |
| UMD | **1.000** | 0.993 | 0.993 | 0.993 | 0.986 | 0.910 | 0.993 |
| UWaveGestureLibraryAll | **0.956** | 0.938 | 0.896 | 0.903 | 0.692 | 0.475 | 0.892 |
| UWaveGestureLibraryX | **0.814** | 0.797 | 0.785 | 0.781 | 0.733 | 0.569 | 0.728 |
| UWaveGestureLibraryY | **0.736** | 0.714 | 0.710 | 0.697 | 0.641 | 0.348 | 0.634 |
| UWaveGestureLibraryZ | 0.749 | **0.759** | 0.757 | 0.721 | 0.690 | 0.655 | 0.658 |
| Wafer | 0.996 | **0.999** | 0.992 | 0.994 | 0.994 | 0.991 | 0.980 |
| Wine | **0.907** | 0.741 | 0.815 | 0.759 | 0.778 | 0.500 | 0.574 |
| WordSynonyms | **0.705** | 0.676 | 0.691 | 0.630 | 0.531 | 0.422 | 0.649 |
| Worms | **0.779** | 0.727 | 0.727 | 0.623 | 0.753 | 0.455 | 0.584 |
| WormsTwoClass | **0.792** | 0.740 | **0.792** | 0.727 | 0.753 | 0.584 | 0.623 |
| Yoga | 0.834 | **0.888** | 0.837 | 0.812 | 0.791 | 0.830 | 0.837 |

