# OpenReview forum: "Ti-MAE: Self-Supervised Masked Time Series Autoencoders"
_ICLR.cc/2023/Conference — Submitted to ICLR 2023_

### Official Review · Reviewer_o7kt · 2022-10-23

**Confidence:** 4
**Clarity, Quality, Novelty And Reproducibility:** 1. The authors give a relatively clea…
**Correctness:** 3
**Technical Novelty And Significance:** 2
**Empirical Novelty And Significance:** 3
**Recommendation:** 5

**Strength And Weaknesses:**

Strength
1. The authors clearly presents the idea of using masked time series autoencoders for time series forecasting and classification.
2. They also did extensive experiments on multiplier real datasets and compared the proposed method with SOTA ones.

Weaknesses
1. "input time series are assumed to follow an integrate distribution"
What does integrate distribution mean?
2. In (1), the polynomial series and the Fourier series are not orthogonal thus the learning might have no unique solution. How are the number of polynomial orders and the Fourier orders decided?
3. It is not clear why 'randomly masks out parts of embedded time series data' can help 'alleviate the distribution shift problem'. Why periodic masking cannot address the same issue? Why do we need randomness here?
4. The way the time series data is randomly masked will affect the performance and is worth being discussed.
5. How can the proposed Ti-MAE make forecasting 'for multiple time windows with various sizes without re-training'? This claim in Section 1 is not justified neither structurally or experimentally.
6. The input time series and the output one has the same dimensionality. This is a quite strong but not necessary assumption. Can the proposed model be applied to forecasting time series with different dimensionality?
7. The ablation study is not complete. Why does data augmentation give worse results? How the way that masks are randomly sampled (as mentioned in 4) affect the performance?
8. It will be helpful to show the training and testing errors and the generalization gap between them.


**Summary Of The Paper:**

This paper proposes a self-supervised framework for time series representation learning. It randomly masks out tokenized time series and learns an autoencoder to reconstruct them at the point-level. The proposed model is applied to time series forecasting and clasification.



**Summary Of The Review:**

The idea is marginally novel and the applications on time series forecasting and classification are interesting.
There are some claims not well justified and need to be addressed.


------------
I have read the authors' responses and would like to keep the same score.

---

> ### Author Response · Authors · 2022-11-18
> **Response to Reviewer o7kt (1/2)**
>
> Thank you for your detailed review and suggestions for improvements. We are glad that you enjoyed reading the paper. Here are our responses to the issues raised.
>
> > **Q1: What does "input sequence follows integrate distribution" mean?**
>
> R: As shown in Figure 1(b) (at Sec.1 of revised submission), through the random masking strategy, the proposed Ti-MAE randomly samples entries in the sequence to form the input sequence and the out sequence for self-supervised tasks, which breaks the rules used in other transformer models that take the sequence following the input sequence as the supervision signal (as demonstrated in Figure 1(a)).  In this way, Ti-MAE better models the integrated distribution of the sequence.
>
> > **Q2: In (1), the polynomial series and the Fourier series are not orthogonal thus the learning might have no unique solution. How are the number of polynomial orders and the Fourier orders decided?**
>
> R: Thank you for pointing out this problem. Equation (1) is an example illustrating the disentanglement of trend and seasonality. In detail, treating the original time series as the combination of the polynomial series and the Fourier series is a simple way to understand why directly utilizing Fourier Transform to capture seasonal features is not effective. The trend part, which is denoted as the polynomial series, seriously affects the capture of periodic features. That is why many existing methods applied disentanglement or difference on input before fed into extractors. (e.g. for a linear trend $x$, we can totally eliminate it by disentanglement or difference.)
>
> > **Q3 \& Q4: It is not clear why 'randomly masks out parts of embedded time series data' can help 'alleviate the distribution shift problem'. Why periodic masking cannot address the same issue? Why do we need randomness here? The way the time series data is randomly masked will affect the performance and is worth being discussed.**
>
> R: Thank you for highlighting this issue. As we mentioned in the original paper and Q1, traditional end-to-end Transformer-based methods can be seen as a special continuous masking strategy, which masks future time series and reconstructs them. The problem of continuous masking strategy is that the encoder of the model can only learn useful information from unmasked historical time series, limiting the utilization of the whole data. When the distribution of historical and future time series diverge, the accuracy of forecasting will degrade. To better demonstrate it, we have added more experiments on different masking strategies, which show the effectiveness of random masking.
>
> |Masking Strategy|Random|Continuous|Split|Periodic
> |:-:|:-:|:-:|:-:|:-:
> |MSE|0.2103|0.3834|0.3564|0.2720
> |MAE|0.2696|0.4420|0.3936|0.3357
>
> Specifically, split masking means we both mask historical time series to reconstruct future values (forecast), and mask future time series to reconstruct historical sequence (lookback); periodic masking means tokens are periodically masked. Notably, periodic masked tokens with a length of four are sampled equidistantly to maintain the same masking ratio. We can see that random masking achieves the best result because randomness can adequately exploit the whole time series data with less inductive bias.
>
> > **Q5: How can the proposed Ti-MAE make forecasting 'for multiple time windows with various sizes without re-training'? This claim in Section 1 is not justified neither structurally or experimentally.**
>
> R: Ti-MAE follows the SSL training paradigm: masking inputs and reconstructing them in the decoder. Due to the consistency between SSL training and downstream forecasting tasks, we can directly generate prediction by the trained decoder. For Ti-MAE, we can manually set different masking ratio after pre-training to generate future time series to be predicted with different lengths (i.e. 50\% masking ratio for an input time series means giving one half of the entries to predict the other half.) Required by other reviewers, we also add the forecasting results after re-training/fine-tune for different settings.

---

> > ### Author Response · Authors · 2022-11-18
> > **Response to Reviewer o7kt**
> >
> > Answers to other clarifications are below:
> > > **Q6: The input time series and the output one has the same dimensionality. This is a quite strong but not necessary assumption. Can the proposed model be applied to forecasting time series with different dimensionality?**
> >
> > R: Sure. Actually the final linear projection layer in the decoder can easily project the input dimensionality to the desired out dimensionality. Here we show the results of using multivariate time series to predict the last univariate target.
> >
> > |||Ti-MAE||CoST||TS2Vec||TNC||MoCo|&emsp;
> > |:-:|:-:|:-:|:-:|:-:|:-:|:-:|:-:|:-:|:-:|:-:|:-:
> > ||Metric|MSE|MAE|MSE|MAE|MSE|MAE|MSE|MAE|MSE|MAE|
> > |ETTh|24|**0.031**|**0.134**|0.040|0.152|0.039|0.151|0.057|0.184|0.040|0.151
> > ||48|**0.048**|**0.167**|0.060|0.186|0.062|0.189|0.094|0.239|0.063|0.191
> > ||168|**0.076**|**0.207**|0.097|0.236|0.142|0.291|0.171|0.329|0.122|0.268
> > ||336|**0.097**|**0.240**|0.112|0.306|0.160|0.316|0.192|0.357|0.144|0.297
> > |Weather|24|**0.005**|**0.056**|0.096|0.213|0.096|0.215|0.102|0.221|0.097|0.216
> > ||48|**0.012**|**0.087**|0.138|0.262|0.140|0.264|0.139|0.264|0.140|0.264
> > ||168|**0.014**|**0.108**|0.207|0.334|0.207|0.335|0.198|0.328|0.198|0.326
> > ||336|**0.015**|**0.115**|0.230|0.356|0.231|0.360|0.215|0.347|0.220|0.350
> >
> > > **Q7: The ablation study is not complete. Why does data augmentation give worse results? How the way that masks are randomly sampled (as mentioned in 4) affect the performance?**
> >
> > R: Data augmentation is widely used in time series for contrastive SSL frameworks to generate different views. However, the ways of augmentation used in literature are borrowed from computer vision or NLP domains, which do not meet the transform-invariant features required by time series data. Forecasting tasks need more fine-grained features, but extra augmentation generally introduces inevitable distortion to the original time series. We thus show the forecasting results with different data augmentation.
> >
> > As for masking and sampling strategies, we have added a group of experiments in the response to  Q4. It is important to emphasize that seeking better sampling strategies is still a general problem in SSL. Meanwhile, proper data augmentation on time series is also a remained open research problem.
> >
> > > **Q8: It will be helpful to show the training and testing errors and the generalization gap between them.**
> >
> > R: Thank you for your insight. But in time series forecasting tasks, we generally need to train one or two epochs on each dataset, which means the training and testing error curves may not be available. Here we present the training and testing errors after one epoch on Weather dataset with different masking strategy.
> >
> > |Masking|Random|Continuous
> > |:-:|:-:|:-:
> > |Training error|0.195|0.356
> > |Testing error|0.165|0.383
> >
> > > **Q9: There are many unclear details, e.g, how the parameters of encoder and decoder are specified.**
> >
> > R: Thank you for your suggestion. Indeed, we did attach most of the details in the supplementary materials. For better reading experience, we have attached the appendix to our revised submission and clarified all mentioned details as much as possible.

---

> > > ### Comment · Reviewer_o7kt · 2022-11-28
> > > **Keep the same score.**
> > >
> > > Thanks for your response and I appreciate all your replies.
> > > For Q1, Q3 and Q4, it is worth exploring how randomness affects the performance. Besides, how the sampling ratio affects the performance? Is the sampling ratio the same for all 4 different masking strategies in your reply?
> > >
> > > For Q2, the idea of combining the polynomial series and the Fourier series is very intuitive but ad-hoc. It is worth demonstrating how removing one of them will affect the autoencoder's performance.
> > >
> > > For Q7, traditional data augmentation does not benefit time series forecasting mainly due to the fact that the transformation in traditional data augmentation changes the temporal correlation in time series data. This should be clearly explained in the manuscript.
> > >
> > > For Q8, it is still not clear why 'training and testing error curves may not be available'. Also, I am a bit concerned that the testing error is smaller than the training error for random settings but larger for the continuous one. This means one is overfitting and the other is underfitting.
> > >
> > > I would like to keep the same score.

---

> > > > ### Author Response · Authors · 2022-11-29
> > > > **Response to Reviewer o7kt (phase 2)**
> > > >
> > > > Thank you for your feedback. Here are our responses to the issues remained.
> > > >
> > > > > **Q1, Q3, Q4: How the sampling ratio affects the performance? Is the sampling ratio the same for all 4 different masking strategies in your reply?**
> > > >
> > > > If the "sampling ratio" refers to "masking ratio", we have attached a group of experiments about masking ratio on different datasets in revised submission (see Appendix B.1). Seeking better masking and sampling strategies is an open problem in self-supervised learning. According to research on the MaskedAE[1] in the vision domain, a masking ratio of 70\% to 90\% is a good choice. Given the continuous nature of time series data shared with images or videos, intuitively a high masking ratio is meaningful. We can see that the best masking ratio is generally around 75\%. Thus, we adopt the 75\% masking ratio across all datasets and have achieved a good performance. The table below summarizes the forecasting results (96-96 setting) with different masking ratio. For different masking strategies in our reply, we have kept the same masking ratio, 75\%. Notably, periodic masked tokens with a length of four are sampled equidistantly to maintain the same masking ratio. We believe that random masking achieves the best result because randomness can adequately exploit the whole time series data with less inductive bias.
> > > >
> > > > ||ETTh||Weather||Exchange||ILI||
> > > > |:-:|:-:|:-:|:-:|:-:|:-:|:-:|:-:|:-:
> > > > |Masking ratio|MSE | MAE | MSE | MAE | MSE | MAE | MSE | MAE
> > > > |30\%|0.6181|0.4984|0.4320|0.4698|0.2707|0.3531|2.2039|0.9908
> > > > |45\%|0.5140|0.4830|0.3082|0.3557|0.2328|0.3380|2.0452|0.9843
> > > > |60\%|0.5011|0.4490|0.2650|0.3414|0.2239|0.3340|2.0389|0.9707
> > > > |75\%|**0.4403**|**0.4338**|**0.2103**|**0.2696**|**0.1701**|**0.2972**|**2.0150**|0.9646
> > > > |90\%|0.4597|0.4385|0.2483|0.3176|0.1952|0.3172|2.0332|**0.9607**
> > > >
> > > > > **Q2: The idea of combining the polynomial series and the Fourier series is very intuitive but ad-hoc. It is worth demonstrating how removing one of them will affect the autoencoder's performance.**
> > > >
> > > > Actually, combining the polynomial series and the Fourier series is just a simple way to explain why many of existing methods, such as FEDformer, Autoformer and CoST, apply disentanglement in the data processing stage. Specifically, FEDformer uses Wavelet Transform and Autoformer uses autocorrelation as a substitute for attention to extract temporal interaction (i.e. seasonality features). Similarly, CoST introduced Fourier Transform to generate new contrastive views in frequency domain for better forecasting performance. These methods all depend on two properties: continuity and stationary (i.e. no trend part) of time series. If we maintain the trend part or destroy the ordering of sequence, the performance of FEDformer, Autoformer and CoST will **drastically degrade**. As for Ti-MAE, we **do not** rely on continuity or stationary of time series. Instead, we utilizes **random masking** for utterly exploiting the whole input time series, and learns the position and alignment of sequences in the reconstruction task, which generally destroys the adverse effect of trend in forecasting. It is important to emphasize that we **do not** apply disentanglement in our proposed Ti-MAE. Here we added one experiment to show the results if we remove trend part (i.e. apply disentanglement on Ti-MAE) in forecasting tasks on Weather dataset (96-96 setting). If we just reconstruct seasonality part and use one extra regressor to make up for trend part, it will not bring about significant changes. Respectively train two autoencoders to reconstruct trend and seasonality parts will bring a slight improvement, but it requires more training time.
> > > >
> > > > |Modification|MSE|MAE
> > > > |:-:|:-:|:-:
> > > > |Default|0.2103|0.2696
> > > > |Disentanglement (extra regressor for trend part)|0.2180|0.2721
> > > > |Disentanglement (two autoencoders)|0.2007|0.2510
> > > >
> > > > > **Q7: Traditional data augmentation does not benefit time series forecasting mainly due to the fact that the transformation in traditional data augmentation changes the temporal correlation in time series data. This should be clearly explained in the manuscript.**
> > > >
> > > > Thank you for your suggestion. We will clarify it in the manuscript.
> > > >
> > > > [1] He, Kaiming et al. “Masked Autoencoders Are Scalable Vision Learners.” 2022 IEEE/CVF Conference on Computer Vision and Pattern Recognition (CVPR) (2022): 15979-15988.

---

> > > > > ### Author Response · Authors · 2022-11-29
> > > > > **Response to Reviewer o7kt (phase 2) continued**
> > > > >
> > > > > > **Q8: It is still not clear why 'training and testing error curves may not be available'. Also, I am a bit concerned that the testing error is smaller than the training error for random settings but larger for the continuous one. This means one is overfitting and the other is underfitting.**
> > > > >
> > > > > Indeed,  our proposed model usually converges after one or two epochs of training. Therefore,  it is hard to provide training or testing error curves (which will look like a point or a segment) as other tasks. For better understanding the training process,  we added the total reconstruction and testing errors of Ti-MAE in each iteration (i.e., each batch) during the training stage (96-96 setting). For continuous masking, we reduce the batch size and provide training and testing error in each iteration too.
> > > > >
> > > > > **Random masking**
> > > > >
> > > > > |Iteration|Reconstruction error|Testing error
> > > > > |:-:|:-:|:-:
> > > > > |0|1.0327|0.7018
> > > > > |10|0.8081|0.6308
> > > > > |20|0.5738|0.5330
> > > > > |30|0.5370|0.5134
> > > > > |40|0.4902|0.5163
> > > > > |50|0.5001|0.5030
> > > > > |100|0.4636|0.4830
> > > > > |110|0.4590|0.4783
> > > > > |120|0.4463|0.4419
> > > > > |130|0.4247|0.4377
> > > > > |140|0.4040|0.4206
> > > > > |200|0.3438|0.3930
> > > > > |210|0.3358|0.3825
> > > > > |250|0.2981|0.3340
> > > > > |260|0.2647|0.3134
> > > > > |270|0.2714|0.3201
> > > > > |300|0.2344|0.2717
> > > > > |350|0.1813|0.2209
> > > > > |380|0.1790|0.2127
> > > > > |390|0.1814|0.2145
> > > > >
> > > > > **Continuous masking**
> > > > >
> > > > > |Iteration|Training error|Testing error
> > > > > |:-:|:-:|:-:
> > > > > |0|0.5820|0.4238
> > > > > |10|0.5478|0.4386
> > > > > |20|0.5462|0.4203
> > > > > |30|0.5315|0.4130
> > > > > |40|0.5001|0.3893
> > > > > |50|0.4859|0.3906
> > > > > |100|0.3985|0.3893
> > > > > |110|0.3977|0.3909
> > > > > |120|0.3894|0.3708
> > > > > |130|0.3921|0.3608
> > > > > |140|0.3894|0.3594
> > > > > |150|0.3871|0.3518
> > > > > |200|0.3875|0.3544
> > > > >
> > > > > We answer the concern on the testing error raised by the reviewer as follows.   For random masking, the training error is actually the reconstruction error, which means that the missing entries will be reconstructed based on the neighboring observed entries. However, for the continuous masking, the missing entries (i.e., the future time steps) will be generated by the past input sequence,  which is a harder task than the random masking. Therefore, this does not means "one is overfitting and the other is underfitting".

---

### Official Review · Reviewer_5eEm · 2022-10-24

**Confidence:** 4
**Correctness:** 3
**Technical Novelty And Significance:** 2
**Empirical Novelty And Significance:** 2
**Recommendation:** 5

**Clarity, Quality, Novelty And Reproducibility:**

The idea is clear and reasonable, but novelty is limited. The empirical results would be more convincing to include the STOA results on long sequence forecasting task.

**Strength And Weaknesses:**

Pros:

1. The proposed MAE based self-supervised learning framework is interesting and reasonable. Experimental results also demonstrate improvement on most real-world datasets.

2. The paper is well-organized, and the writing is easy to follow.


Cons:

1. The experiments are not comprehensive. The baselines reported in the current version are not SOTA. How are the comparison results with the recent transformer based models, like FEDformer and ETSformer as the authors mentioned? Note that there are some lightweight models (Dlinear, DeepTIME, etc.) which can beat the transformer based models. It would be more convincing to compare these models.


2. The proposed method is reasonable, but the novelty is limited. It is quite straightforward to apply MAE on time series data. Could the authors elaborate more on the challenges when applying MAE on time series data?

Questions:

1. Does the proposed method follow the pre-training + fine-tuning paradigm? It seems the randomly mask strategy could align the reconstruction and prediction tasks and eliminate the effort to build an extra forecasting model.

2. How is the model complexity compared with other transformed based forecasting models?

**Summary Of The Paper:**

This work propose a simple and novel self-supervised learning framework for time series representation learning. Instead of using contrastive representation learning, this work directly follow the masked data modeling and optimize the reconstruction loss of the randomly masked data points. Empirical studies on real-world datasets and ablation studies show the effectiveness and scalability of the proposed framework.

**Summary Of The Review:**

see above
____

I would like to thank the author's great effort for the detailed response and additional experiments. However, I would like to keep the original score due to the following concerns. First, the novelty of methodology and technique are limited. I appreciate the explanation of MAE and the proposed method, but I still feel the proposed method is a straightforward solution to apply it on time series data.  Second, I understand the point from the author that there is no need to compare works unpublished, like deeptime, dlinear. But there exists some published paper which achieves much better results than the baselines selected in this manuscript, for example [1,2]. It would be promising  to see a significant improvement with a simple method in this time series forecasting benchmark. Third, the proposed method has quadratic complexity which is worse than the $L\log L$ complexity of the baseline transformers.

[1] Learning Latent Seasonal-Trend Representations for Time Series Forecasting
[2] FiLM: Frequency improved Legendre Memory Model for Long-term Time Series Forecasting

---

> ### Author Response · Authors · 2022-11-18
> **Response to Reviewer 5eEm  (1/2)**
>
> Thank you for your valuable comments and feedback. Here are our responses to the issues raised.
>
> > **Q1: The experiments are not comprehensive. The baselines reported in the current version are not SOTA in forecasting tasks.**
>
> R: We would like to emphasize the goal of this paper is to learn effective representations for downstream tasks including forecasting and classification. For fair comparison with other SOTA Transformer-based methods including FEDformer and ETSformer on forecasting, we have fine-tuned Ti-MAE on forecasting tasks. Specifically, we extract the encoder of Ti-MAE and freeze it after pre-training, and add an extra linear regressor for fine-tuning. The updated forecasting results are shown as the following table. It can be observed that our proposed model have achieved remarkably better results on the datasets. In addition, we acknowledge that Dlinear and DeepTime have achieved promising results, however, these two papers have not been published yet. Especially, we notice that Dlinear was rejected by NeurIPS and withdrew from openreview, while DeepTime was under reviewed by ICLR 2023 (parallel to our submission) and was questioned by reviewers for lacking of clarity on motivation and lacking of empirical validation (e.g., unclear lookback length which may cause unfair comparison). Therefore, we believe that it is neither safe nor appropriate to include their results at the current stage.
>
> |Methods||Ti-MAE$\dagger$||ETSformer||FEDformer||Autoformer||Informer|&emsp;
> |:-:|:-:|:-:|:-:|:-:|:-:|:-:|:-:|:-:|:-:|:-:|:-:
> |||MSE|MAE|MSE|MAE|MSE|MAE|MSE|MAE|MSE|MAE
> |ETTh|12|**0.2826**|**0.3383**|0.4479| 0.4582|0.3272|0.3940|0.5016|0.5204|0.4299|0.4644
> ||24|**0.3430**|**0.3816**|0.4602| 0.4621|0.3699|0.4185|0.5063|0.5309|0.4880|0.4963
> ||48|**0.3705**|**0.3939**|0.4855| 0.4735|0.3912|0.4347|0.5703|0.5563|0.6625|0.5774
> ||96|**0.4039**|**0.4074**|0.5090| 0.4851|0.4194|0.4476|0.6052|0.5663|0.9584|0.7157
> ||128|**0.4270**|**0.4208**|0.5279| 0.4949|0.4360|0.4551|0.6043|0.5726|0.9504|0.7197
> ||168|**0.4455**|**0.4363**|0.5446| 0.5044|0.4733|0.4783|0.7382|0.6199|1.1043|0.7867
> |Weather|12 |**0.0811**|**0.1199**|0.0900|0.1537|0.1476|0.2350|0.2042|0.2960|0.2351|0.3128
> ||24 |**0.1065**|**0.1484**|0.1396|0.2224|0.1624|0.2496|0.2200|0.3141|0.1244|0.2022
> ||48 |**0.1290**|**0.1784**|0.1848|0.2735|0.1993|0.2898|0.2691|0.3542|0.2352|0.3129
> ||96 |**0.1633**|**0.2151**|0.2034|0.2994|0.2350|0.3139|0.2891|0.3673|0.2808|0.3586
> ||128|**0.1774**|**0.2283**|0.2092|0.2972|0.2395|0.3148|0.2758|0.3469|0.3055|0.3723
> ||168|**0.2031**|**0.2525**|0.2199|0.3016|0.2632|0.3281|0.2861|0.3506|0.3473|0.4003
> |Exchange|24|0.0276|0.1167|**0.0266**|**0.1130**|0.0717|0.1958|0.0894|0.2239|0.4963|0.5623
> ||48|**0.0438**|0.1481|0.0441|**0.1464**|0.0954|0.2247|0.1474|0.2881|1.0477|0.8169
> ||96|**0.0814**|0.2074|0.0861|**0.2044**|0.1470|0.2790|0.2883|0.3957|1.1038|0.8215
> ||128|**0.1108**|**0.2361**|0.1153|0.2373|0.1886|0.3153|0.3102|0.4107|1.1978|0.8535
> ||168|**0.1443**|**0.2824**|0.1549|0.2773|0.2484|0.3638|0.3066|0.4108|1.1564|0.8444
> ||196|**0.1661**|0.3040|0.1830|**0.3034**|0.2718|0.3800|0.2990|0.4021|1.1679|0.8545
> |ILI|24|**2.4781**|**0.9925** |3.1358|1.2128|3.3017|1.2689|3.3292|1.2088|4.2526|1.3551
> ||36|**2.2103**|**0.8956** |2.9369|1.1218|2.6125|1.0575|3.4076|1.1688|4.7647|1.4433
> ||48|**1.9697**|**0.8826** |2.9386|1.1120|2.5883|1.0683|3.2077|1.1125|4.8189|1.4553
> ||60|**2.3496**|**0.9545** |2.8840|1.1324|2.8460|1.1533|3.3373|1.1659|4.7974|1.4669
> ||72|**2.1563**|**0.8884** |2.8615|1.1579|2.8921|1.1721|3.1079|1.1237|4.1188|1.3718
> ||96|**2.3860**|**0.9827** |3.1109|1.2186|3.1048|1.2412|3.0530|1.1260|4.5218|1.4401

---

> > ### Author Response · Authors · 2022-11-18
> > **Response to Reviewer 5eEm (2/2)**
> >
> > Answers to other clarifications are below:
> > > **Q2: The proposed method is reasonable, but the novelty is limited. It is quite straightforward to apply MAE on time series data. Could the authors elaborate more on the challenges when applying MAE on time series data?**
> >
> > That is a good question. Indeed, our one motivation derives from a key point we noticed: traditional Transformer-based end-to-end methods could be seen as a special continuous masking strategy, which masks future time series and reconstructs them. The problem of continuous masking strategy is that the encoder of the model can only learn useful information from the unmasked historical time series, limiting the utilization of the overall distribution of time series data. Instead, compared with the continuous masking strategy, the random masking strategy proposed in this paper explores the overall distribution of the sequence, leading to an enhancement in the performance on forecasting tasks. This random masking strategy coincides with the idea of the MaskedAE firstly proposed in the image domain, which provides us insights of adopting the masking strategy in the time series domain.
> >
> > However, it is challenging to directly apply the MaskedAE from the vision domain to time series data due to the difference between image and time series. Generally, each natural image is individually collected, which means that we can easily apply MAE on every single image via resizing. However, directly segmenting time series data into different patches (each of which can be regarded as a single image) will result in divergence during the training of the MaskedAE.
> > In practice, we need to empirically adopt a proper segmentation strategy to sample overlapping segments, and apply the MaskedAE on each segment.
> >
> > Notably, the MaskedAE in vision outperforms well in classification tasks, but it is not so good in generation tasks. Thus, we made some adjustment on the setting of hyperparameters and network structure. For instance, we choose to reconstruct a single point rather than a patch (as MaskedAE in vision) to maintain the consistency to forecasting tasks (i.e., predicting future sequence values at the point level). In addition, the number of hidden states of the MaskedAE applied on time series (64 in our experiment) should be less than on images (around 2000 in He et al., 2021) for better training of Ti-MAE.
> >
> > > **Q3: Does the proposed method follow the pre-training + fine-tuning paradigm?**
> >
> > R: This is a good observation. In the first edition, since the decoder can directly generate future time series to be predicted, we reported the forecasting results directly from the pre-trained Ti-MAE without fine-tuning. For fair comparison with other end-to-end models, we have added the new forecasting results of fine-tuned Ti-MAE in revised submission. Notably, our pre-training and fine-tuning are respectively conducted on the same dataset, rather than pre-training on one extra large dataset as in CV/NLP.
> >
> > > **Q4: How is the model complexity compared with other Transformer-based forecasting models?**
> >
> > R: Referring to the reports of Transformer-based models, many models have $O(L\log L)$ complexity, however, there exists a large constant since these methods generally need to do a bulk of pre-treatment (e.g. Fourier Transform, Wavelet Transform), which makes the overall training not that efficient. In comparison, although our proposed Ti-MAE has $O(L^2)$ complexity due to the vanilla attention mechanism, we need to pre-train the encoder of Ti-MAE **only once** and can fine-tune it on different forecasting settings. The runtime analysis (seconds) of Transformer-based models is shown in the following table, where we execute three times for each setting (using 96 historical steps to predict future steps of 24, 48, 96, 288 and 672 respectively) on one single Nvidia V100 GPU. Thus, the total running time of Ti-MAE is less than other Transformer-based methods.
> >
> > |Stage|H|Ti-MAE|FEDformer|ETSformer|Autoformer|Informer
> > |:-:|:-:|:-:|:-:|:-:|:-:|:-:
> > |Pre-training|/|335.1 $\pm$ 0.5|/|/|/|/
> > |Training|24|17.5 $\pm$ 0.2|170.1 $\pm$ 2.8|68.6 $\pm$ 0.7|61.8 $\pm$ 0.4|68.4 $\pm$ 0.7
> > ||48|17.6 $\pm$ 0.1|199.2 $\pm$ 2.1|69.1 $\pm$ 0.6|68.7 $\pm$ 0.7|76.1 $\pm$ 0.6
> > ||96|17.9 $\pm$ 0.2|250.1 $\pm$ 2.0|72.1 $\pm$ 0.5|79.8 $\pm$ 0.4|87.7 $\pm$ 0.2
> > ||288|18.6 $\pm$ 0.2|324.0 $\pm$ 1.8|73.1 $\pm$ 1.0|130.5 $\pm$ 0.9|137.0 $\pm$ 0.3
> > ||672|19.8 $\pm$ 0.3|460.9 $\pm$ 2.1|76.3 $\pm$ 0.6|220.5 $\pm$ 0.4|220.1 $\pm$ 0.2
> > |Inference|24|5.7 $\pm$ 0.1|9.1 $\pm$ 0.4|9.0 $\pm$ 0.4|13.6 $\pm$ 0.2|9.3 $\pm$ 0.1
> > ||48|6.1 $\pm$ 0.2|10.9 $\pm$ 0.4|9.1 $\pm$ 0.3|15.1 $\pm$ 0.2|10.2 $\pm$ 0.2
> > ||96|6.5 $\pm$ 0.1|12.4 $\pm$ 0.1|9.9 $\pm$ 0.3|18.0 $\pm$ 0.3|12.1 $\pm$ 0.5
> > ||288|8.7 $\pm$ 0.7|16.7 $\pm$ 0.4|13.3 $\pm$ 0.5|31.3 $\pm$ 0.5|18.8 $\pm$ 0.3
> > ||672|12.9 $\pm$ 1.1|25.2 $\pm$ 1.3|19.1 $\pm$ 1.4|55.6 $\pm$ 0.9|31.4 $\pm$ 1.2

---

### Official Review · Reviewer_jXSb · 2022-10-25

**Confidence:** 3
**Correctness:** 3
**Technical Novelty And Significance:** 2
**Empirical Novelty And Significance:** 2
**Recommendation:** 5

**Clarity, Quality, Novelty And Reproducibility:**

This paper is pretty clear. The figures are easy to read. The novelty is marginal. It looks reproducible.

**Strength And Weaknesses:**

Strengths:
1. It's novel to propose the masked time series auto encoders which can learn strong representations with less inductive bias or hierarchical trick.
2. The experiments are evaluated on various of settings, e.g. multivariate time series forecasting, classification


Weaknesses:
1. The masking ratio is tricky. It highly depends on the dataset. When dataset changes, the masking ratio may change. The paper didn't mention how to deal with this issue.
2. The experiments do not compare with some state-of-the-art method, such as NHITS, FedFormer and etc.
3. No ablation study on the effect of different backbone models, e.g. LSTM, Transformer, TCN and etc.

**Summary Of The Paper:**

This paper propose a masked time series autoencoders which can learn strong representations with less inductive bias or hierarchical trick. The proposed Ti-MAE bridges the connection between contrastive representation learning and generative Transformer-based methods. Ti-MAE adequately leverages all the input sequence and alleviates the distribution shift problem. The flexible setting of masking ratio makes Ti-MAE more adaptive to various prediction scenarios with different time steps. Experimental results demonstrate the efficacy of proposed method.

**Summary Of The Review:**

This paper proposes Ti-MAE to bridge the connection between contrastive representation learning and generative Transformer-based methods. It can learn strong representations with less inductive bias or hierarchical trick. Ti-MAE adequately leverages all the input sequence and alleviates the distribution shift problem. The flexible setting of masking ratio makes Ti-MAE more adaptive to various prediction scenarios with different time steps. Experimental results demonstrate the efficacy of proposed method.

However, the masking ratio is tricky. It highly depends on the dataset. When dataset changes, the masking ratio may change. The paper didn't mention how to deal with this issue. The experiments do not compare with some state-of-the-art method, such as NHITS, FedFormer and etc. No ablation study on the effect of different backbone models, e.g. LSTM, Transformer, TCN and etc.

---

> ### Author Response · Authors · 2022-11-18
> **Response to Reviewer jXSb (1/2)**
>
> Thank you for your insightful review and comments. Here are our responses to the issues raised:
>
> > **Q1: The masking ratio is tricky. It highly depends on the dataset. When dataset changes, the masking ratio may change.**
>
> R: This is a great insight. Seeking better masking and sampling strategies is an open problem in self-supervised learning. According to research on the MaskedAE[1] in the vision domain, a masking ratio of 70\% to 90\% is a good choice. Given the continuous nature of time series data shared with images or videos, intuitively a high masking ratio is meaningful. To verify this point, here we present the forecasting results on different datasets with different masking ratios as follows. We can see that the best masking ratio is generally around 75\%. Thus, we adopt the 75\% masking ratio across all datasets and have achieved a good performance.
>
> ||ETTh||Weather||Exchange||ILI||
> |:-:|:-:|:-:|:-:|:-:|:-:|:-:|:-:|:-:
> |Masking ratio|MSE | MAE | MSE | MAE | MSE | MAE | MSE | MAE
> |30\%|0.6181|0.4984|0.4320|0.4698|0.2707|0.3531|2.2039|0.9908
> |45\%|0.5140|0.4830|0.3082|0.3557|0.2328|0.3380|2.0452|0.9843
> |60\%|0.5011|0.4490|0.2650|0.3414|0.2239|0.3340|2.0389|0.9707
> |75\%|**0.4403**|**0.4338**|**0.2103**|**0.2696**|**0.1701**|**0.2972**|**2.0150**|0.9646
> |90\%|0.4597|0.4385|0.2483|0.3176|0.1952|0.3172|2.0332|**0.9607**
>
>
> > **Q2: The experiments do not compare with some state-of-the-art method in forecasting tasks.**
>
> R: We would like to emphasize that the goal of this paper is to learn effective representations for downstream tasks including classification and forecasting. Indeed, it is hard to directly compare the performance between representation learning methods and supervised learning methods. For fair comparison, we have fine-tuned Ti-MAE on forecasting tasks. Specifically, we extract the encoder of Ti-MAE and freeze it after pre-training, and then add an extra linear regressor for fine-tuning. We include the updated results and the comparison with two recent SOTA Transformer-based methods, i.e., ETSformer[2] and FEDformer[3], as shown in the following table. It can be observed that our proposed model have achieved remarkably better results on the datasets. Note that the code URL (https://bit.ly/3JLIBp8) of N-HITS provided in the arxiv (https://arxiv.org/abs/2201.12886v5) is not accessible.
>
> |Methods||Ti-MAE$\dagger$||ETSformer||FEDformer||Autoformer||Informer|&emsp;
> |:-:|:-:|:-:|:-:|:-:|:-:|:-:|:-:|:-:|:-:|:-:|:-:
> |||MSE|MAE|MSE|MAE|MSE|MAE|MSE|MAE|MSE|MAE
> |ETTh|12|**0.2826**|**0.3383**|0.4479| 0.4582|0.3272|0.3940|0.5016|0.5204|0.4299|0.4644
> ||24|**0.3430**|**0.3816**|0.4602| 0.4621|0.3699|0.4185|0.5063|0.5309|0.4880|0.4963
> ||48|**0.3705**|**0.3939**|0.4855| 0.4735|0.3912|0.4347|0.5703|0.5563|0.6625|0.5774
> ||96|**0.4039**|**0.4074**|0.5090| 0.4851|0.4194|0.4476|0.6052|0.5663|0.9584|0.7157
> ||128|**0.4270**|**0.4208**|0.5279| 0.4949|0.4360|0.4551|0.6043|0.5726|0.9504|0.7197
> ||168|**0.4455**|**0.4363**|0.5446| 0.5044|0.4733|0.4783|0.7382|0.6199|1.1043|0.7867
> |Weather|12 |**0.0811**|**0.1199**|0.0900|0.1537|0.1476|0.2350|0.2042|0.2960|0.2351|0.3128
> ||24 |**0.1065**|**0.1484**|0.1396|0.2224|0.1624|0.2496|0.2200|0.3141|0.1244|0.2022
> ||48 |**0.1290**|**0.1784**|0.1848|0.2735|0.1993|0.2898|0.2691|0.3542|0.2352|0.3129
> ||96 |**0.1633**|**0.2151**|0.2034|0.2994|0.2350|0.3139|0.2891|0.3673|0.2808|0.3586
> ||128|**0.1774**|**0.2283**|0.2092|0.2972|0.2395|0.3148|0.2758|0.3469|0.3055|0.3723
> ||168|**0.2031**|**0.2525**|0.2199|0.3016|0.2632|0.3281|0.2861|0.3506|0.3473|0.4003
> |Exchange|24|0.0276|0.1167|**0.0266**|**0.1130**|0.0717|0.1958|0.0894|0.2239|0.4963|0.5623
> ||48|**0.0438**|0.1481|0.0441|**0.1464**|0.0954|0.2247|0.1474|0.2881|1.0477|0.8169
> ||96|**0.0814**|0.2074|0.0861|**0.2044**|0.1470|0.2790|0.2883|0.3957|1.1038|0.8215
> ||128|**0.1108**|**0.2361**|0.1153|0.2373|0.1886|0.3153|0.3102|0.4107|1.1978|0.8535
> ||168|**0.1443**|**0.2824**|0.1549|0.2773|0.2484|0.3638|0.3066|0.4108|1.1564|0.8444
> ||196|**0.1661**|0.3040|0.1830|**0.3034**|0.2718|0.3800|0.2990|0.4021|1.1679|0.8545
> |ILI|24|**2.4781**|**0.9925** |3.1358|1.2128|3.3017|1.2689|3.3292|1.2088|4.2526|1.3551
> ||36|**2.2103**|**0.8956** |2.9369|1.1218|2.6125|1.0575|3.4076|1.1688|4.7647|1.4433
> ||48|**1.9697**|**0.8826** |2.9386|1.1120|2.5883|1.0683|3.2077|1.1125|4.8189|1.4553
> ||60|**2.3496**|**0.9545** |2.8840|1.1324|2.8460|1.1533|3.3373|1.1659|4.7974|1.4669
> ||72|**2.1563**|**0.8884** |2.8615|1.1579|2.8921|1.1721|3.1079|1.1237|4.1188|1.3718
> ||96|**2.3860**|**0.9827** |3.1109|1.2186|3.1048|1.2412|3.0530|1.1260|4.5218|1.4401
>
> [1] He, Kaiming et al. “Masked Autoencoders Are Scalable Vision Learners.” 2022 IEEE/CVF Conference on Computer Vision and Pattern Recognition (CVPR) (2022): 15979-15988.
>
> [2] Woo, Gerald et al. “ETSformer: Exponential Smoothing Transformers for Time-series Forecasting.” arXiv preprint arXiv:2202.01381 (2022).
>
> [3] Zhou, Tian et al. “FEDformer: Frequency Enhanced Decomposed Transformer for Long-term Series Forecasting.”  NIPS (2022).

---

> > ### Author Response · Authors · 2022-11-18
> > **Response to Reviewer jXSb (2/2)**
> >
> > Answers to other clarifications are below:
> >
> > > **Q3: No ablation study on the effect of different backbone models.**
> >
> > R: Thank you for your suggestion. We have added the ablation study on the effect of different backbone (i.e. TCN[1], LSTM[2]) in supplementary materials before (Appendix B.3). For better reading experience, we have attached all of them to our revised submission. Here we also added the results of different backbone on classification (UCR) and forecasting tasks (on the Weather dataset under the 200-100 setting).
> >
> > |Backbone|Classification|Forecasting|&emsp;
> > |:-:|:-:|:-:|:-:
> > ||Avg.Acc|MSE|MAE
> > |Transformer|**0.8231**|**0.2103**|**0.2696**
> > |TCN|0.7729 (-6.1\%)|0.2773 (-31.9\%)|0.3343 (-24.0\%)
> > |LSTM|0.6834 (-17.0\%)|0.2840 (-35.0\%)|0.3401 (-26.1\%)
> >
> > [1] Bai, Shaojie et al. “An Empirical Evaluation of Generic Convolutional and Recurrent Networks for Sequence Modeling.” arXiv preprint arXiv:1803.01271 (2018).
> >
> > [2] Hochreiter, Sepp and Jürgen Schmidhuber. “Long Short-Term Memory.” Neural Computation 9 (1997): 1735-1780.

---

### Official Review · Reviewer_6CpA · 2022-10-27

**Confidence:** 4
**Correctness:** 3
**Technical Novelty And Significance:** 2
**Empirical Novelty And Significance:** 3
**Recommendation:** 6

**Clarity, Quality, Novelty And Reproducibility:**

This work adapts an existing model to timeseries data. The work is of high-quality and clarity.

**Strength And Weaknesses:**

S

- solid experimental work in terms of datasets and models compared
- good results with multiple ablation tests

W

- lack of connection between motivation (disentanglement) and the actual proposed method
- non-significant result compared to TS2Vec baseline (figure 6)



**Summary Of The Paper:**

This paper proposes an adaptation of the popular MaskedAE model for timeseries data.

**Summary Of The Review:**

I am positive about the contribution of this paper which might be considered a bit incremental, however, it's the first work in this area. I am reluctant about the comparison with TS2Vec in Figure 6 which shows that both methods are identical.

---

> ### Author Response · Authors · 2022-11-18
> **Response to Reviewer 6CpA**
>
> Thank you for the positive review and feedback. Here are our responses to the issues raised:
>
> > **Q1: Lack of connection between motivation (disentanglement) and the actual proposed method (Ti-MAE).**
>
> R: Thank you for spotting this issue, we have made the appropriate changes of writing and illustration (Sec. 1 and Fig. 1 in revised submission). We briefly clarify the connection between our motivation and the proposed method as follows.
>
> In time series data, it is crucial to model the consistency between the observed sequences and the forecasting sequence. An intuitive approach widely observed in Transformer-based methods to force the consistency is to disentangle the trend from the time series data. However, these forecasting models deal with time series data in a way of receiving historical time series as the input and then making prediction for future time, which are indeed insufficient in exploring of the distribution of the whole sequence.
> To demonstrate this point more clearly, we can regard this way used in existing Transformer-based methods as a special **continuous masking strategy** (as shown in Figure 1 (a)), which masks future time series and reconstructs them. However,  in the continuous masking strategy, the encoder of the model can only learn useful information from unmasked historical time series, limiting the exploration and utilization of the whole data distribution. Instead, our proposed Ti-MAE utilizes **random masking** (as shown in Figure 1 (b)) rather than the continuous masking for utterly exploiting the whole input time series, and learns the position and alignment of sequences in the reconstruction task, which generally destroys the adverse effect of trend in forecasting. For verification, we add a result of comparison between random masking and continuous masking on the Weather dataset under the 96-96 setting, as shown in the following table.
>
> |Masking Strategy|MSE|MAE
> |:-:|:-:|:-:
> |Random|**0.2103**|**0.2696**
> |Continuous|0.3834|0.4420
>
> We also add a group of transfer experiments to demonstrate the robustness of Ti-MAE when coping with time series with different trend parts in the appendix B.5.
>
> > **Q2: Non-significant result in classification tasks compared to TS2Vec baseline.**
>
> R: It is important to note that the goal of this paper is to learn effective representations for downstream tasks including classification and forecasting.   Here we list the results of Ti-MAE and TS2Vec on forecasting and classification tasks. Ti-MAE outperforms on forecasting tasks and achieves competitive performance with TS2Vec. Notably, as an incremental work of TS2Vec, CoST added contrastive learning on the frequency domain for getting a better performance on forecasting tasks, while impairing the classification performance ( from 0.8201 to 0.7872). Thus it is very challenging to perform well on both the classification task and the forecasting task. Instead, Ti-MAE delivers competitive classification results, while significantly outperform these contrastive learning methods in the forecasting task.
>
> |Model|Classification (UCR)|Forecasting (Weather)
> |:-:|:-:|:-:
> |Ti-MAE|**0.8231** (+0.36\%)|**0.1633** (+60.4\%)
> |TS2Vec|0.8201|0.4289
> |CoST|0.7872|0.4119
>
> [1] Yue, Zhihan et al. “TS2Vec: Towards Universal Representation of Time Series.” AAAI (2022).
>
> [2] Woo, Gerald et al. “CoST: Contrastive Learning of Disentangled Seasonal-Trend Representations for Time Series Forecasting.” ICLR (2022)

---

### Author Response · Authors · 2022-12-09
**General Response To All Reviewers**

We thank all the reviewers for the time and effort in reviewing our paper and providing constructive suggestions. This has certainly improved our paper. We summarize our major efforts during this phase as follows:
* We clarified the motivations of our proposed Ti-MAE with random masking strategy.
* We performed further forecasting experiments on the requested SOTA Transformer-based models.
* We performed further ablation studies about: 1. the effect of different backbone models (e.g. Transformer, TCN, LSTM), 2. the impact of masking ratio on different datasets, 3. the impact of different masking strategies.
* We provided the training time and computation resources analysis.
* We provided a group of forecasting results with different input and output dimension.
* We modified some expressions in the original paper to improve clarity, and corrected typos in the paper.

We hope that our response addresses your concerns and comments. Since we are reaching the end of the discussion period, if there are any further questions or comments we can address, please let us know!

---

### Decision · Program_Chairs · 2023-01-20

**Decision:**

Reject

**Justification For Why Not Higher Score:**

Although this paper has an average score of 5.25, I still don't consider it as a borderline paper and recommend a rejection for two main reasons. First, majority of the reviewers (3 out of 4) have negative ratings. Second, I agree with two of the reviewers that this paper is a straightforward application of MAE on time series data and thus the novelty is limited.

**Justification For Why Not Lower Score:**

N/A

**Metareview: Summary, Strengths And Weaknesses:**

This paper proposed a MAE based self-supervised learning framework for time-series representation learning. A random masking strategy was adopted on the time-series data and then the authors learned the model by optimizing the reconstruction loss of those randomly masked data points.

Strengths:
1. The paper is clearly written.
2. The model performance on time series forecasting is very promising.
3. The experiments are quite comprehensive. After the rebuttal, it is very nice that the authors added experiments to show the effect of different backbones and the effect of the masking ratio. The authors also included additional baselines as suggested by the reviewers.

Weaknesses:
1. I agree with two reviewers (6CpA and 5eEm) that this work is a straightforward application of MAE on time series data. The novelty is not sufficient for a top-conference like ICLR.
2. Performance improvement on classification is marginal. To make this model general for time series analytics (forecasting, classification and regression), the authors should address challenges specific to time series. Following the weakness above, additional modules to address these challenges will improve the novelty.
3. Although the current experiments are comprehensive, one reviewer (5eEm) still raised a valid point that the baselines reported in the paper are not SOTA. The authors can include the comparison with FiLM, LaST,  DeepTime, DLinear and NHITS in the future version.